# MYBL2 and ATM suppress replication stress in pluripotent stem cells

Daniel Blakemore[1], Nuria Vilaplana-Lopera[1] (ID), Ruba Almaghrabi[1], Elena Gonzalez[1], Miriam Moya[1], Carl Ward[2,3], George Murphy[4], Agnieszka Gambus[1] (ID), Eva Petermann[1] (ID), Grant S Stewart[1,†] (ID) & Paloma García[1,*,†] (ID)

## Abstract

**Replication stress, a major cause of genome instability in cycling cells, is mainly prevented by the ATR-dependent replication stress response pathway in somatic cells. However, the replication stress response pathway in embryonic stem cells (ESCs) may be different due to alterations in cell cycle phase length. The transcription factor MYBL2, which is implicated in cell cycle regulation, is expressed a hundred to a thousand-fold more in ESCs compared with somatic cells. Here we show that MYBL2 activates ATM and suppresses replication stress in ESCs. Consequently, loss of MYBL2 or inhibition of ATM or Mre11 in ESCs results in replication fork slowing, increased fork stalling and elevated origin firing. Additionally, we demonstrate that inhibition of CDC7 activity rescues replication stress induced by MYBL2 loss and ATM inhibition, suggesting that uncontrolled new origin firing may underlie the replication stress phenotype resulting from loss/inhibition of MYBL2 and ATM. Overall, our study proposes that in addition to ATR, a MYBL2-MRN-ATM replication stress response pathway functions in ESCs to control DNA replication initiation and prevent genome instability.**

**Keywords** B-MYB; DNA damage; ESCs; iPSC; origin firing

**Subject Categories** DNA Replication, Recombination & Repair; Stem Cells & Regenerative Medicine

# Introduction

DNA replication is a highly complex process that requires tight regulation to ensure that genome stability is maintained. Obstacles to DNA replication activate the replication stress response pathway, which not only functions to ensure that the initiation of DNA replication and progression through the cell cycle progression are suppressed, but also acts to facilitate the repair and restart of damaged replication forks (Bartek *et al*, 2004; Aguilera & Gomez-Gonzalez, 2008; Zeman & Cimprich, 2014).

Two main serine/threonine protein kinases are responsible for controlling the cellular response to genetic damage and impediments to DNA replication: ataxia-telangiectasia mutated (ATM) and ataxia-telangiectasia and Rad3 related (ATR). ATR is activated following its recruitment to RPA coated ssDNA, a common intermediate that occurs during normal DNA replication and also as a consequence of replication stress. It has been shown that ATR is essential for the survival of proliferating cells as its loss is embryonic lethal (de Klein *et al*, 2000). In contrast, ATM is recruited by the MRN complex to DNA ends and is primarily associated with signalling the presence of DNA double-strand breaks (DSB) (Uziel *et al*, 2003; Lee & Paull, 2005). Activation of the ATR kinase during replication leads to the phosphorylation and activation of the CHK1 kinase, which functions to suppress new replication origin firing (Guo *et al*, 2000; Liu *et al*, 2000; Moiseeva *et al*, 2019), promote fork stability and prevent premature entry into mitosis (Cimprich & Cortez, 2008). In contrast, it is thought that ATM is activated during S-phase only upon MRN-dependent recruitment to sites of DSBs, i.e. collapsed replication forks (Bakkenist & Kastan, 2003; Lee & Paull, 2004). However, various studies have challenged this canonical pathway for ATM activation by demonstrating that its recruitment and activity is dependent upon chromatin context, indicating that ATM may be capable of being activated in the absence of DSBs (Bakkenist & Kastan, 2003; Ewald *et al*, 2008; Bencokova *et al*, 2009; Cam *et al*, 2010; Olcina *et al*, 2010; Olcina *et al*, 2013; Iwahori *et al*, 2014). Furthermore, studies performed in *Xenopus laevis* egg extracts have indicated a specific role for ATM in regulating the timing of replication (Marheineke & Hyrien, 2004; Shechter *et al*, 2004). Therefore, it has been hypothesized that ATM could have additional roles in replication control, specifically in non-somatic cell types.

In embryonic stem cells (ESCs), safeguarding genome stability during DNA replication is of extreme importance as alterations to

---

1 Institute of Cancer and Genomic Science, College of Medical and Dental Sciences, University of Birmingham, Birmingham, UK
2 Laboratory of Integrative Biology, Guangzhou Institutes of Biomedicine and Health, Chinese Academy of Sciences (CAS), Guangzhou, China
3 Chinese Academy of Sciences (CAS), Key Laboratory of Regenerative Biology and Guangdong Provincial Key Laboratory of Stem Cell and regenerative medicine, Guangzhou Institutes of Biomedicine and Health, Guangzhou, China
4 Department of Medicine, Boston University School of Medicine, Boston, MA, USA
  *Corresponding author. Tel: +44 0 121 414 4093; E-mail: p.garcia@bham.ac.uk
  †These authors contributed equally to this work as senior authors

the genome will be transmitted to their differentiated daughter cells during development, potentially compromising tissue integrity and function. Human and mouse ESCs proliferate very rapidly and possess an atypical cell cycle with short GAP phases and a weak G1-S checkpoint (Savatier *et al,* 1994; Ballabeni *et al,* 2011; Coronado *et al,* 2013; Kapinas *et al,* 2013). It has been suggested that mouse embryonic stem cells (mESCs) may contain a significant fraction of unreplicated DNA as they enter mitosis (Ahuja *et al,* 2016). Nevertheless, despite these traits, pluripotent stem cells actually have a very low mutation rate (Cervantes *et al,* 2002; Fujii-Yamamoto *et al,* 2005; Kapinas *et al,* 2013) and maintain genome stability more efficiently than somatic cell types (Murga *et al,* 2007; Kapinas *et al,* 2013; Zhao *et al,* 2015; Ahuja *et al,* 2016). However, how these cells retain such a low mutational rate in the presence of high levels of endogenous replication stress is still not properly understood.

One protein that is unusually highly expressed in human and mouse ESCs and key for genome stability is the transcription factor MYBL2 (alias, B-MYB) (Sitzmann *et al,* 1996; Tarasov *et al,* 2008; Lorvellec *et al,* 2010). Similar to ATR, targeted disruption of MYBL2 leads to early embryonic lethality (Tanaka *et al,* 1999). MYBL2 has been shown to be important for transactivating the promoters of genes responsible for regulating the G2/M transition (Osterloh *et al,* 2007; Tarasov *et al,* 2008; Knight *et al,* 2009; Lorvellec *et al,* 2010; Wolter *et al,* 2017). More specifically, the MuvB core cooperates with MYBL2 to recruit FOXM1 to the promoters of specific genes, such as Cyclin B and survivin the CDC25 phosphatases, which are responsible for mediating G2–M checkpoint control (Lefebvre *et al,* 2010; Martinez & Dimaio, 2011; Down *et al,* 2012; Sadasivam *et al,* 2012; Sadasivam & DeCaprio, 2013). In addition to this, a recent study identified a signalling axis, involving ATR-CDK1-FOXM1 and demonstrated that it cooperates with MYBL2 to govern the proper exit from S-phase into G2 (Saldivar *et al,* 2018). Moreover, it has also been shown that YAP, a component of the HIPPO signalling pathway, interacts with the MYBL2-MuvB complex to influence the expression of genes important for mitosis (Pattschull *et al,* 2019). Whilst the ability of MYBL2 to act as a transcription factor is critical for cell cycle regulation, our previous work in mouse ESCs has highlighted the importance of MYBL2 for maintaining normal replication fork progression under unperturbed conditions, as stem cells lacking MYBL2 displayed a significant reduction in replication fork speed (Lorvellec *et al,* 2010). However, whether this replication phenotype is caused by dysfunction of the ATR-CDK1-FOXM1 axis or aberrant regulation of YAP-dependent gene transcription remains unknown. Here, we demonstrate that the replication stress caused by loss of MYBL2 in mouse ESCs is epistatic with loss of Mre11 or ATM function and can be rescued by suppressing origin firing via inhibition of CDC7. This suggests that uncontrolled replication initiation underlies the replication stress phenotype exhibited by cells lacking MYBL2 and that it is not caused by alterations in the expression of genes involved in controlling the cell cycle. Taken together, this work identifies ATM as a critical regulator of origin firing in pluripotent stem cells and highlights the importance of MYBL2-ATM in controlling the initiation of origin firing and the replication stress response in these cells.

# Results

## $Mybl2^{\Delta/\Delta}$ ESCs display a replication stress phenotype characterized by a decrease in replication fork speed and an increase in replication fork instability

Consistent with our previous work (Lorvellec *et al,* 2010), DNA fibre analysis revealed that replication fork progression was significantly reduced in $Mybl2^{\Delta/\Delta}$ ESCs compared to a WT counterpart, with a 65% reduction in the median replication fork speed (Figs 1A and EV1A). Analysis of the relative percentage of different replication structures revealed a notable decrease in the percentage of elongating forks in the $Mybl2^{\Delta/\Delta}$ ESCs compared to $Mybl2^{+/+}$ ESCs (Fig EV1B and C). This reduction in actively elongating replication fork structures was associated with an increase in fork stalling and new origin firing, the latter of which is a known compensation mechanism invoked to counteract a reduction in the cellular replicative capacity (Fig EV1D and E).

To further investigate potential changes in the kinetics of replication fork progression, the ratio of first and second label replication

---

**Figure 1.  $Mybl2^{\Delta/\Delta}$ ESCs display fork instability and a replication stress phenotype leading to unreplicated DNA, Increase replication-associated DSB and genome instability.**

A  Distribution curve of replication fork rates for $Mybl2^{+/+}$ and $Mybl2^{\Delta/\Delta}$ ESCs. Statistical analysis was performed using the Mann–Whitney *U*-test. *n* = 6 experimental replicates. A minimum of 490 replication forks were counted (****$P < 0.0001$).

B  Left panel: representative elongating replication forks (stable and unstable). Quantification of the average IdU/CldU ratio in the $Mybl2^{+/+}$ and $Mybl2^{\Delta/\Delta}$ ESCs. Percentage of highly unstable forks (ratio above 2) are indicated. *Y*-axis was cut to change scale and display large values. Statistical analysis was performed using the Mann–Whitney *U*-test (****$P < 0.0001$). At least 450 ratios were recorded from 6 experimental replicates.

C  The length of both CldU labels surrounding IdU tracks (newly fired origins) was measured before calculating a ratio representing symmetry around new origins. Quantification of the average positive ratio in the $Mybl2^{+/+}$ and $Mybl2^{\Delta/\Delta}$ ESCs. *Y*-axis was cut to change scale and display large values. Statistical analysis was performed using the Mann–Whitney *U*-test. 55 and 75 ratios were recorded in the $Mybl2^{+/+}$ and $Mybl2^{\Delta/\Delta}$, respectively, from three replicates (****$P < 0.0001$).

D  Representative image showing the presence of UFB by positive Immunostaining for PICH. (Scale bar 10 μm). Frequency of anaphases positive for UFBs relative to all anaphases in the $Mybl2^{+/+}$ and $Mybl2^{\Delta/\Delta}$ ESCs. At least 75 anaphases were counted per experimental group from three independent experiments. Statistical analysis was performed using an unpaired two-tailed *t*-test (****$P < 0.0001$).

E  Representative images for comets in $Mybl2^{+/+}$ and $Mybl2^{\Delta/\Delta}$ untreated and $Mybl2^{+/+}$ and $Mybl2^{\Delta/\Delta}$ plus CPT (20× magnification). (Scale bar 50 μm).

F  Quantification of the mean and distributions of olive tail moments in $Mybl2^{+/+}$ and $Mybl2^{\Delta/\Delta}$ untreated and CPT-treated. *Y*-axis was cut to change scale and display large values. Statistical analysis using Mann–Whitney *U*-tests. At least 300 comets were measured for each group from 5 experimental replicates (****$P < 0.0001$).

G  Frequency of EdU-positive cells with over 6 53BP1 foci in $Mybl2^{+/+}$, $Mybl2^{\Delta/\Delta}$ untreated and $Mybl2^{+/+}$ and $Mybl2^{\Delta/\Delta}$ plus CPT. Error bars represent SEM. Statistical analysis using unpaired two-tailed *t*-test. At least 200 EdU-positive nuclei were analysed from 4 experimental repeats.

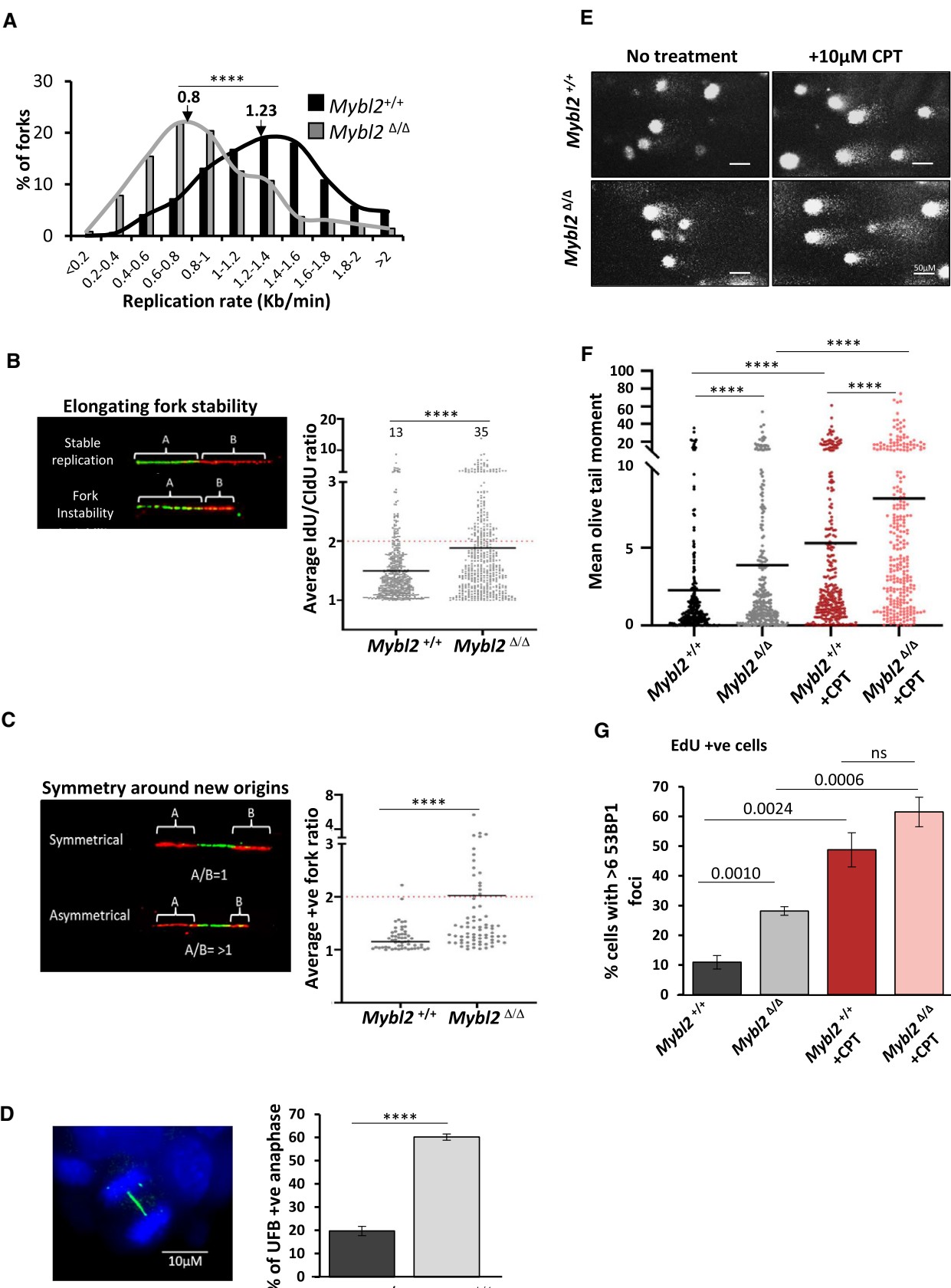

**Figure 1.**

tracks of elongating forks was calculated as a measure of replication fork stability (Maya-Mendoza *et al*, 2018). An increase in the ratio with respect to wild-type cells suggests a disruption of continuous fork progression due to increased fork instability. Notably, the *Mybl2*$^{\Delta/\Delta}$ ESCs exhibited a significant increase in replication fork instability compared to *Mybl2*$^{+/+}$ ESCs (Fig 1B). In keeping with this observation, *Mybl2*$^{\Delta/\Delta}$ ESCs also displayed a strong asymmetry phenotype of newly fired replication forks initiating from the same origin when compared to *Mybl2*$^{+/+}$ ESCs (Fig 1C). This indicates that MYBL2 plays a critical role in maintaining replication fork stability in the absence of exposure to any exogenous genotoxins.

It is known that prolonged replication stress leads to the formation of ultra-fine bridges (UFBs) between separating sister chromatids in anaphase due to the presence of under-replicated DNA (Chan *et al*, 2018). To investigate whether loss of MYBL2 leads to increased UFBs, immunofluorescence staining using an antibody to PICH, a DNA translocase known to coat ultra-fine bridges during anaphase (Baumann *et al*, 2007; Chan & Hickson, 2011) was carried out on asynchronous *Mybl2*$^{+/+}$ and *Mybl2*$^{\Delta/\Delta}$ ESCs. In the *Mybl2*$^{+/+}$ ESCs, 20% of anaphases exhibited UFBs, which is comparable to levels previously observed for normal stem cells (Broderick *et al*, 2015; Hengeveld *et al*, 2015; Saldivar *et al*, 2018). Strikingly, there was a significant increase (60%) in the percentage of UFB-positive anaphases in the *Mybl2*$^{\Delta/\Delta}$ compared to *Mybl2*$^{+/+}$ ESCs (Fig 1D). Overall, these data build upon our previous findings and strongly demonstrate that MYBL2 loss in ESCs culminates in chronic replication stress leading to under-replicated DNA passing into mitosis.

## *Mybl2*$^{\Delta/\Delta}$ ESCs exhibit increased replication-associated genome instability

Cells under chronic replication stress exhibit elevated levels of DNA damage that can be partly attributed to an increase in stalled forks, which if left unresolved, are vulnerable to collapse into double-strand breaks (DSBs) (Petermann *et al*, 2010; Zeman & Cimprich, 2014). Therefore, to determine whether the increased replication stress present in MYBL2 deficient ESCs resulted in more replication-associated DNA breakage, total DNA breaks were quantified using an alkaline comet assay (Ostling & Johanson, 1984; Singh *et al*, 1988). Exposure of cells to camptothecin (CPT), a genotoxic agent known to induce replication-associated DNA breakage, was used as a positive control. Notably, the *Mybl2*$^{\Delta/\Delta}$ ESCs exhibited significantly increased levels of spontaneous DNA breakage as measured by the average olive tail moment (OTM) when compared to the

*Mybl2*$^{+/+}$ ESCs (Fig 1E and F). Whilst the exposure of ESCs to CPT elevated the DNA breakage to a level that was not significantly different between the two genotypes.

To ascertain whether the increase in OTM in *Mybl2*$^{\Delta/\Delta}$ ESCs was due to DSB formation during replication, immunofluorescence was performed using 53BP1 and EdU as markers of DSBs and cells in S-phase, respectively. Again, exposure to CPT was used as a positive control for replication-associated DNA breakage. We observed a twofold increase in the percentage of S-phase nuclei displaying more than six 53BP1 foci in *Mybl2*$^{\Delta/\Delta}$ cells (EdU positive) when compared to the *Mybl2*$^{+/+}$ ESCs (Fig 1G and Appendix Fig S1). As expected, in response to CPT treatment, both the *Mybl2*$^{+/+}$ and *Mybl2*$^{\Delta/\Delta}$ ESCs exhibited a large increase in the percentage of cells with over six 53BP1 foci. These findings suggest that the increased spontaneous DNA damage in the *Mybl2*$^{\Delta/\Delta}$ ESCs most likely arises as a consequence of replication fork collapse.

## *Mybl2*$^{\Delta/\Delta}$ ESCs fail to activate the DNA damage-activated, G2/M cell cycle checkpoint

Given that *Mybl2*$^{\Delta/\Delta}$ ESCs show signs of chronic replication stress (Fig 1A–C) leading to unreplicated DNA (Fig 1D) and DNA breakage (Fig 1E and F), it would be expected that these cells would arrest in S- or G2-phase of the cell cycle due to prolonged activation of the DNA damage checkpoint response (Zeman & Cimprich, 2014). Interestingly however, *Mybl2*$^{\Delta/\Delta}$ ESCs still retain the capacity to proliferate even in the presence of DNA damage. Since MYBL2 has been shown to regulate the transcription of genes linked with cell cycle regulation (Katzen *et al*, 1998; Tarasov *et al*, 2008; Sadasivam *et al*, 2012; Henrich *et al*, 2017; Saldivar *et al*, 2018), we hypothesized that the capacity of the *Mybl2*$^{\Delta/\Delta}$ cells to continue cycling in the presence of replication stress and DNA damage could be attributed to deregulation of the DNA damage cell cycle checkpoints. To investigate this, *Mybl2*$^{+/+}$ and *Mybl2*$^{\Delta/\Delta}$ cells were treated with CPT and then DNA fibre analysis was used to monitor the DNA damage-induced suppression of new origin firing as a marker of activation of the intra-S-phase checkpoint. Unexpectedly, we did not observe a reduction in new origin firing following exposure to CPT in *Mybl2*$^{+/+}$ ESCs, rather, the level of new origin firing increased. In ESCs lacking MYBL2, CPT treatment had no observable effect on the already high level of new origin firing (Fig 2A). This suggests that like the G1/S-phase checkpoint, murine ESCs do not retain the capacity to activate the DNA damage-induced intra-S-phase checkpoint and that the observed increase in new origin firing following exposure to

**Figure 2. MYBL2-ablated ESCs fail to activate the DNA-damaged activated, G2/M checkpoint.**

A   Cells were treated for 1.5 h with 5 μM CPT, before sequential addition of IdU and CldU for 20 min each. Frequency of new firing origins (relative to total structures counted) from *Mybl2*$^{+/+}$ and *Mybl2*$^{\Delta/\Delta}$ ESCs treated or not with CPT ($n = 3$ independent experiments; error bars indicate SEM). Statistical analysis using two-tailed unpaired *t*-test (ns = no significant).

B   Distribution of replication rate in *Mybl2*$^{+/+}$ and *Mybl2*$^{\Delta/\Delta}$ ESC treated or not with CPT as above. Statistical analysis was carried out using unpaired Mann–Whitney *U*-tests. A minimum of 200 forks were quantified for each of the genotypes and conditions shown from three independent repeats ($*P < 0.05$; $**P < 0.01$; $***P < 0.001$; $****P < 0.0001$).

C   ESCs were treated with or without CPT for 4 h before 0.1 μg/ml colcemid was added during the final 3.5 h of treatment. Representative image of H3-pS10-positive cells (turquoise) (Scale bar 10 μm). Frequency of H3-pS10-positive cells in *Mybl2*$^{+/+}$ and *Mybl2*$^{\Delta/\Delta}$ ESCs. Data from three independent experiments. Error bars represent SEM. Statistical analysis was carried out using a two-tailed unpaired *t*-test ($*P < 0.05$; $**P < 0.01$; ns = no significant).

D   P-CDK1 (Tyr15), CDK1 and Beta-actin expression levels of *Mybl2*$^{+/+}$ and *Mybl2*$^{\Delta/\Delta}$ ESCs with or without CPT treatment analysed by immunoblotting. Bar graph represents average band density of P-CDK1 in *Mybl2*$^{\Delta/\Delta}$ ESCs relative to loading control and relative to the MYBL2$^{+/+}$ from six repeats. Error bars represent SEM. Statistical analysis was carried out using a two-tailed unpaired *t*-test ($**P < 0.01$).

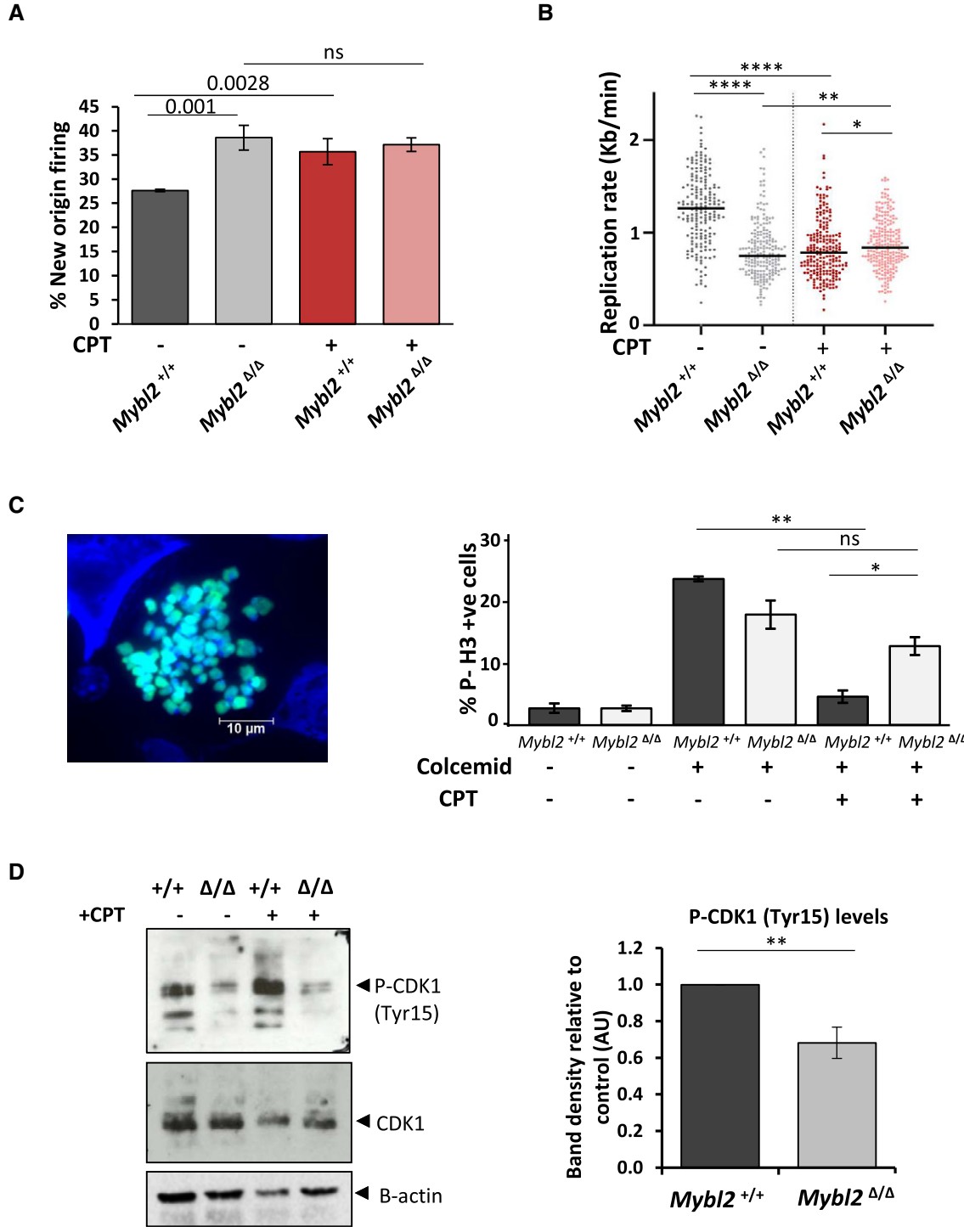

**Figure 2.**

CPT represents an adaptive response of the cell to trigger more origin firing when faced with high levels of replication stress. In agreement with this, CPT treatment resulted in a significant slowing of DNA replication in *Mybl2*⁺/⁺ ESCs, which was comparable to the replication rate of untreated *Mybl2*^Δ/Δ ESCs (Fig 2B). Interestingly, the slow rate of replication of the *Mybl2*^Δ/Δ ESCs was unaffected by exposure to CPT, indicating that the compromised rate of replication

is sufficient to prevent any significant collisions between replication machinery and CPT-induced abortive Top1 complexes.

To assess whether ESCs were also incapable of activating the G2/M DNA damage checkpoint, *Mybl2*⁺/⁺ and *Mybl2*^Δ/Δ ESCs were treated with CPT and then the percentage of cells passing into mitosis in the presence of DNA damage was quantified using histone H3 phosphorylated on serine-10 (H3-pS10) as a marker of mitotic cells.

To prevent cells traversing through mitosis, cells were treated also with colcemid, a tubulin depolymerizing agent which arrests cells in metaphase (Fig 2C) (Li *et al*, 2005). As expected, robust activation of the G2/M checkpoint by CPT was induced in the $Mybl2^{+/+}$ ESCs as measured by a significant reduction in H3-pS10 positive cells (from 23.3 to 4.5%). Interestingly, exposure of $Mybl2^{\Delta/\Delta}$ ESCs to CPT failed to induce a significant reduction in H3-pS10 positive cells, indicative of an inability to activate the DNA damage-induced G2/M checkpoint (Fig 2C). These data are consistent with the role of MYBL2 in regulating the G2-M transition in the presence of DNA damage.

A critical event facilitating the ability of cells to activate the G2/M checkpoint is the WEE1-dependent phosphorylation of CDK1 on tyrosine-15 (Tyr-15) (Heald *et al*, 1993; Watanabe *et al*, 1995). Therefore, to ascertain whether the G2/M checkpoint defect observed in the $Mybl2^{\Delta/\Delta}$ ESCs was caused by aberrant regulation of CDK1, we utilized Western blotting to measure the levels of CDK1 Tyr-15 phosphorylation. This analysis revealed that $Mybl2^{\Delta/\Delta}$ ESCs exhibited reduced levels of CDK1 phosphorylated on Tyr-15 when compared to $Mybl2^{+/+}$ ESCs, whilst no obvious variations in CDK1 protein levels were observed (Fig 2D). As expected, inhibition of the WEE1 kinase completely abolished phospho-CDK1 levels in both the $Mybl2^{+/+}$ and $Mybl2^{\Delta/\Delta}$ ESCs (Appendix Fig S2). To analyse the effect of additional stress upon CDK1 activity in the $Mybl2^{\Delta/\Delta}$ ESCs, cells were treated with CPT for 4 h. Following CPT treatment, $Mybl2^{+/+}$ ESCs displayed an increase in the P-CDK1 (Tyr15) levels, presumably reflecting the inhibition of CDK1 activity by cell cycle checkpoint signalling. In contrast, CPT-treated $Mybl2^{\Delta/\Delta}$ ESCs did not display any obvious increase in P-CDK1, consistent with these cells lacking this DNA damage-activated checkpoint response (Fig 2D). Overall, these data indicate that chronic loss of MYBL2 in ESCs results in abnormal CDK1 activity leading to a weaker G2/M cell cycle checkpoint.

### Reduced CHK1 activation in MYBL2-ablated ESCs after exposure to DNA damage

To investigate the underlying cause of the deregulated CDK1 activity in $Mybl2^{\Delta/\Delta}$ cells, we focused on CHK1, which is the principal checkpoint kinase acting downstream of ATR and is known to be important for the suppression of origin firing and cell cycle checkpoint activation through its ability to inhibit CDK1 and CDK2 (Petermann *et al*, 2010). Western blotting analysis of CHK1 in

unperturbed conditions showed that the level of phospho-CHK1 (Ser345) in the $Mybl2^{\Delta/\Delta}$ ESCs was slightly increased compared to $Mybl2^{+/+}$ ESCs (Fig 3A), which is consistent with the increased spontaneous replication stress we have observed in these cells. However, the level of P-CHK1 in the $Mybl2^{\Delta/\Delta}$ ESCs was significantly reduced when compared to $Mybl2^{+/+}$ ESCs following exposure to CPT, suggesting that $Mybl2^{\Delta/\Delta}$ ESCs can not efficiently activate CHK1 in response to certain types of genotoxic stress (Fig 3A).

To investigate further the mechanism of CHK1 activation in ESCs, cells were treated with either an ATR (AZ20) or ATM (KU60019) inhibitor for 2 h before subjecting the cells to 5 Gy ionizing radiation to induce DNA damage. In line with our previous observation, $Mybl2^{\Delta/\Delta}$ ESCs failed to efficiently phosphorylate CHK1 after exposure to IR as compared to the WT ESCs (Fig 3B). Furthermore, treatment with the ATR inhibitor reduced the levels of P-CHK1 in the ESCs irrespective of genotype (Fig 3B). Interestingly, treatment with the ATM inhibitor also appeared to reduce the levels of P-CHK1 in the $Mybl2^{+/+}$ but not the $Mybl2^{\Delta/\Delta}$, indicating not only the presence of crosstalk between ATR and ATM signalling pathways with respect to CHK1 activation in ESCs but also that a loss of MYBL2 compromises the ability of ATM to be activated (Fig 3B). Moreover, the increased levels of spontaneous P-CHK1 observed in the untreated $Mybl2^{\Delta/\Delta}$ ESCs were also reduced by the ATR inhibitor but not the ATM inhibitor (Appendix Fig S3). Altogether, our data suggest that CHK1 activation is dependent on both ATR and ATM signalling and that MYBL2 may function upstream of ATM to activate CHK1.

### Inhibition of ATR signalling results in severe replication stress independently of MYBL2

$Mybl2^{\Delta/\Delta}$ ESCs are unable to properly activate CHK1 in response to exogenous DNA damage, suggesting a potential role for MYBL2 in regulating the replication stress response downstream of ATM and possibly ATR. To determine whether MYBL2 acts in one or both of these replication stress response pathways, we utilized the ATR and ATM inhibitors to ascertain whether the endogenous replication stress observed in the $Mybl2^{\Delta/\Delta}$ ESCs was epistatic with inhibition of the activity of either one of these kinases.

Initially, $Mybl2^{+/+}$ and $Mybl2^{\Delta/\Delta}$ ESCs were treated with an ATR inhibitor for 1.5 h before labelling active replication and performing DNA fibre analysis (Fig EV2A). In both the $Mybl2^{+/+}$ and $Mybl2^{\Delta/\Delta}$ ESCs, ATR inhibition resulted in a highly significant decrease in

**Figure 3. Proficient ATR-dependent CHK1 activation in MYBL2-ablated ESC.**

A   Immunoblot showing the levels of phosphorylated CHK1 (Ser345) and GAPDH in the $Mybl2^{+/+}$ and $Mybl2^{\Delta/\Delta}$ ESCs with or without CPT treatment (2.5 µM CPT for 4 h).

B   Immunoblot showing the levels of P-CHK1 (Ser345) and CHK1 in irradiated $Mybl2^{+/+}$, and $Mybl2^{\Delta/\Delta}$ ESCs with or without 3 h inhibitor treatment: ATR (AZ20, 5 µM) and ATM (Ku60019, 10 µM). Cells were also exposed to 5 Gy irradiation before the final hour of inhibitor treatment. Bar graph (lower panel) represents average band density of P-CHK1 in $Mybl2^{\Delta/\Delta}$ ESCs relative to CHK1 and relative to untreated cells. $Mybl2^{+/+}$ from two independent repeats and $Mybl2^{\Delta/\Delta}$ ESCs from three independent repeats. Error bars represent SD. Statistical analysis was carried out using a two-tailed unpaired *t*-test (*$P < 0.05$).

C   Cells were treated for 1.5 h with 5 µM AZ20, before sequential addition of IdU and CldU for 20 min each. Dot plot representing the effect of ATR inhibition upon replication rate in $Mybl2^{+/+}$ and $Mybl2^{\Delta/\Delta}$ ESC. Statistical analysis of distributions was carried out using unpaired Mann–Whitney *U*-tests. A minimum of 240 forks was quantified for each of the genotypes and conditions shown from two independent repeats (****$P < 0.0001$)

D   Frequency of new firing origins (relative to total structures counted) from $Mybl2^{+/+}$ and $Mybl2^{\Delta/\Delta}$ ESCs treated or not with ATR inhibitor AZ20. At least 500 replication structures were counted per treatment from two independent repeats. Statistical analysis using two-tailed unpaired *t*-test (*$P < 0.05$; ns = no significant).

E   Distribution and average of IdU/CldU fork ratio in $Mybl2^{+/+}$ and $Mybl2^{\Delta/\Delta}$ ESCs with or without ATR inhibitor treatment. Percentage of highly unstable forks (ratio above 2) are indicated. Dotted line at 1 indicates positive ratio. Y-axis was cut to change scale and display large values. Statistical analysis was carried out using the Mann–Whitney *U*-test. At least 150 ratios were calculated per treatment group from two independent repeats (*$P < 0.05$; **$P < 0.01$; ***$P < 0.001$; ****$P < 0.0001$).

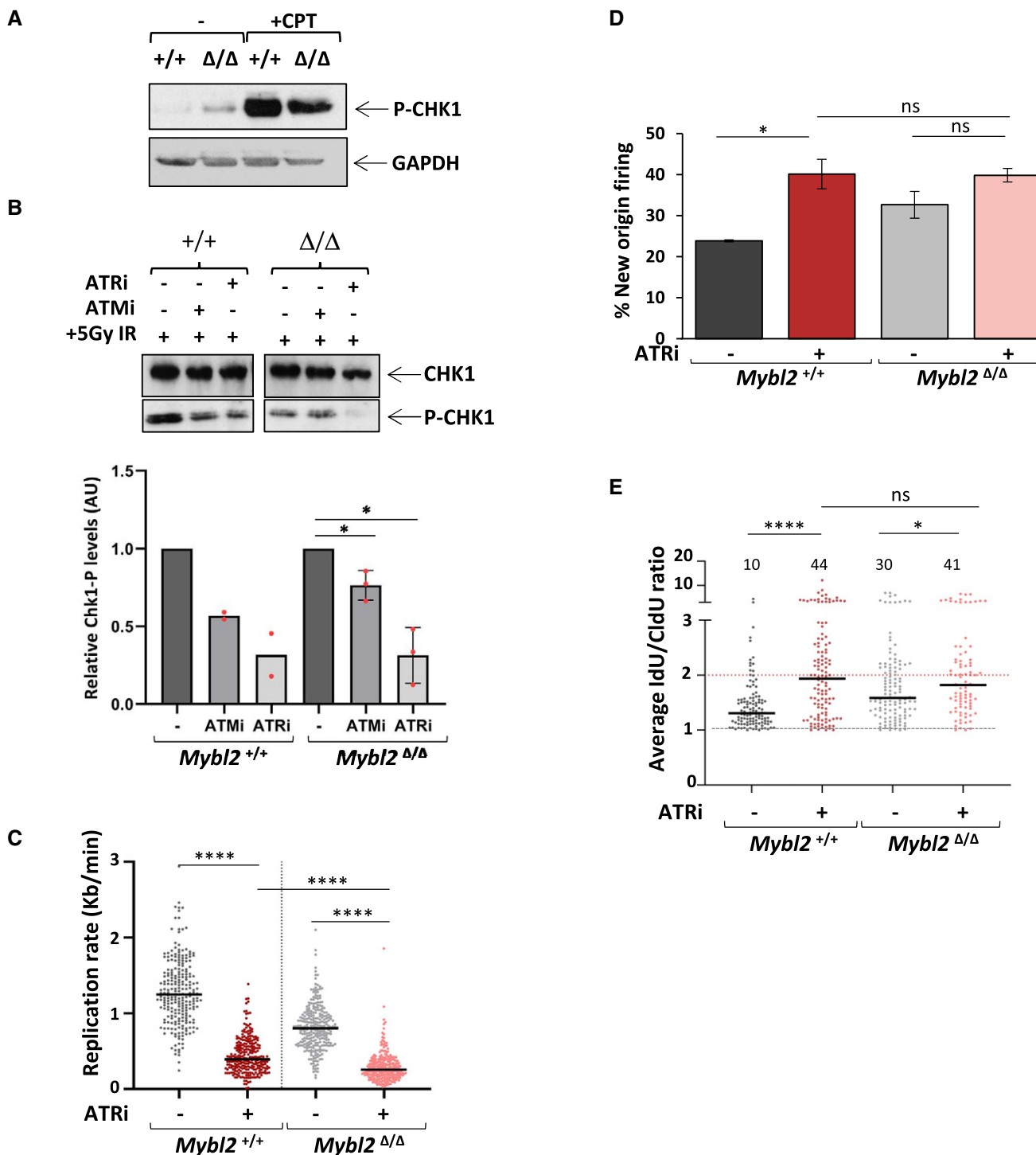

**Figure 3.**

replication fork progression (Figs 3C and, EV2B and C). However, inhibition of ATR only deregulated replication initiation in the *Mybl2*$^{+/+}$ but not the *Mybl2*$^{\Delta/\Delta}$ ESCs (Figs 3D and EV2D). To determine whether ATR inhibition affected fork stability, the ratio between the lengths of first- and second-labelled tracks of elongating forks was calculated. The average fork ratio in the *Mybl2*$^{+/+}$ ESCs treated with ATR inhibitor increased significantly when compared

to that of untreated cells (Figs 3E and EV2D), whereas in contrast, fork instability only moderately increased in *Mybl2*$^{\Delta/\Delta}$ ESCs following treatment with the ATR inhibitor (Figs 3E and EV2D). Overall, these findings suggest that MYBL2 may function downstream of ATR to regulate the response to replication stress in ESCs but that loss of ATR has a much greater impact on replication fork stability and elongation rates than the loss of MYBL2.

## ATM and MYBL2 function together to prevent replication stress and genome instability in pluripotent stem cells

Given the severity of the replication stress phenotype induced following ATR inhibition relative to that present in the $Mybl2^{\Delta/\Delta}$ ESCs, it is conceivable that either MYBL2 only functions within a subset of ATR-dependent responses to replication stress or acts within a parallel pathway that facilitates the ATR-dependent replication stress response, i.e. a pathway regulated by ATM. Therefore, to investigate a link between MYBL2 and ATM, $Mybl2^{+/+}$, and $Mybl2^{\Delta/\Delta}$ ESCs were treated with a chemical inhibitor of ATM before the addition of IdU and CldU to label actively replicating DNA (Fig 4A). DNA fibre analysis revealed that replication fork progression in the $Mybl2^{+/+}$ ESCs was surprisingly sensitive to ATM inhibition, albeit not as sensitive to ATR inhibition, with the median replication forks rate decreasing to levels similar to $Mybl2^{\Delta/\Delta}$ ESCs (Fig 4B). Interestingly, ATM inhibition in the $Mybl2^{\Delta/\Delta}$ ESCs had no additional effect upon replication forks progression rate (Fig 4B) indicating that the spontaneous replication stress present in the $Mybl2^{\Delta/\Delta}$ ESCs most likely arises as a consequence of a compromised ATM-dependent replication stress response pathway. Consistent with ATM and MYBL2 functioning in the same pathway, inhibition of ATM led to a significant increase in replication fork asymmetry and new origin firing in $Mybl2^{+/+}$ ESCs but not in the $Mybl2^{\Delta/\Delta}$ ESCs (Fig 4C and D), which was not due to any alterations in cell cycle profile (Fig EV3). However, in keeping with ATR playing a predominant role in controlling the replication stress response in ESCs and ATM facilitating this, combined inhibition of ATR and ATM reduced replication rates in ESCs irrespective of genotype to levels comparable to those observed following inhibition of ATR alone (Fig EV2E).

To confirm that the treatment with ATM inhibitor was causing replication stress similar to that seen in the $Mybl2^{\Delta/\Delta}$ ESCs, the formation of UFBs following exposure to the ATM inhibitor was assessed. As expected, the $Mybl2^{+/+}$ ESCs treated with ATM inhibitor accumulated under-replicated DNA, as evidenced by the high percentage of cells displaying UFB-positive anaphases (70%). Notably, ATM inhibition induced UFBs in $Mybl2^{+/+}$ ESCs at a level comparable to that observed in $Mybl2^{\Delta/\Delta}$ ESCs (Fig 4E). Next, we asked the question whether the DNA damage in the $Mybl2^{\Delta/\Delta}$ ESCs is mimicked by ATM inhibition in WT ESCs. To test this, $Mybl2^{+/+}$ and $Mybl2^{\Delta/\Delta}$ ESCs were treated with an ATM inhibitor before performing immunofluorescence staining for 53BP1 in cells also treated with EdU (Fig EV4A). Inhibition of ATM in the $Mybl2^{+/+}$ ESCs resulted in an increase in 53BP1 foci in EdU-positive cells with a threefold increase in the percentage of cells with more than six foci (Fig EV4B and C). In contrast, ATM inhibition in the $Mybl2^{\Delta/\Delta}$ ESCs had no additional effect upon the already elevated levels of 53BP1 recruitment (Fig EV4B and C). Together, these data suggest that MYBL2 functions upstream of ATM in ESCs to suppress replication stress and genome instability. Based on this, we treated $Mybl2^{+/+}$ and $Mybl2^{\Delta/\Delta}$ ESCs with CPT and used Western blotting to directly monitor the activation of ATM using a phospho-specific antibody to S1987; a validated marker of ATM autoactivation. Strikingly, $Mybl2^{\Delta/\Delta}$ ESCs were unable to efficiently activate ATM following the induction of DNA damage as compared to the $Mybl2^{+/+}$ ESCs (Fig 4F). Additionally, inhibition of CHK2 activity, a direct target of ATM (Matsuoka *et al*, 2000), in the $Mybl2^{\Delta/\Delta}$ ESCs had no additional effect on replication fork rates (Appendix Fig S4A). These observations serve to strength the notion that MYBL2 is stem cell-specific activator of the ATM-dependent replication stress response.

To verify our findings suggesting a role for ATM in regulating the replication stress response in ESCs without using small molecule inhibitors, we generated $Atm^{-/-}$ ESCs and used DNA fibre analysis to monitor replication stress. This analysis demonstrated that genetic loss of ATM gave rise to a similar reduction in replication fork speed to that which we had previously observed in the $Mybl2^{+/+}$ ESCs treated with an ATM inhibitor (Fig 4G). Importantly, this reduced replication speed was not further reduced by treating the $Atm^{-/-}$ ESC with an ATM inhibitor, demonstrating that the

---

**Figure 4. ATM kinase and MYBL2 function to regulate replication in pluripotent stem cells.**

A   Scheme of the protocol. ESCs were treated with 10 μM KU60019 for 1.5 h before sequential addition of IdU and CldU for 20 min each. DNA was treated and visualized as previously described.

B   Distribution of replication rate after treatment with/without ATM inhibitor. Statistical analysis of distributions was carried out using an unpaired Mann–Whitney U-test. $n = 4$. A minimum of 270 forks was measured for each condition (****$P < 0.0001$).

C   Distribution of the average IdU/CldU ratio in $Mybl2^{+/+}$ and $Mybl2^{\Delta/\Delta}$ ESCs with or without ATM inhibitor treatment. Percentage of highly unstable forks (ratio above 2) are indicated. Dotted line at 1 indicates positive ratio. Y-axis was cut to change scale and display large values. Statistical analysis was carried out using the Mann–Whitney U-test. At least 260 ratios were calculated per treatment group from four separate experiments (****$P < 0.0001$).

D   Frequency of new fired origins for $Mybl2^{+/+}$ and $Mybl2^{\Delta/\Delta}$ ESCs with or without ATM inhibitor treatment. Error bars represent SEM. Statistical analysis was carried out using unpaired t-test. At least 500 replication structures were counted per treatment group from four separate repeats.

E   Frequency of anaphases positive for UFB positive cells (based on ERCC/PICH-positive immunostaining) for $Mybl2^{+/+}$, the $Mybl2^{\Delta/\Delta}$ and the $Mybl2^{+/+}$ treated for 2 h with ATM inhibitor (KU60019). Data represent at least 100 anaphases from each group from two experimental repeats. Statistical analysis was performed using an unpaired two-tailed t-test.

F   Immunoblot showing the levels of phosphorylated ATM (Ser1987) and ATM in the $Mybl2^{+/+}$ and $Mybl2^{\Delta/\Delta}$ ESCs with or without CPT treatment (10 μM CPT for 2 h). Bar graph (lower panel) represents average band density of P-ATM in $Mybl2^{\Delta/\Delta}$ ESCs treated with CPT relative to ATM and relative to wild-type CPT-treated cells from three independent repeats. Error bars represent SD. Statistical analysis was carried out using a two-tailed unpaired t-test.

G   Distribution plot of replication speed in $Mybl2^{+/+}$ and $ATM^{-/-}$ ESC treated or not for 90 min with 10 μM KU60019 (ATMi), before sequential addition of IdU and CldU for 20 min each. Statistical analysis was carried out using the Mann–Whitney U-test. At least 200 replication forks were counted from 3 separate experiments for wild type and $ATM^{-/-}$ and two separate experiments for $ATM^{-/-}$ treated with ATM inhibitor (****$P < 0.0001$).

H   Quantification of the number of replication factories per cell for the different genotypes and treatments. A minimum of 65 early S-phase nuclei was counted per condition from two independent repeats. Statistical analysis was performed using a two-tailed unpaired unequal variance t-test (****$P < 0.0001$).

I   Distribution plot of replication speed in human iPSC and MEFs (primary and immortal). Statistical analysis was carried out using the Mann–Whitney U-test. At least 400 replication forks for iPSC, 200 for primary MEFS and 250 for immortal MEFs were counted from 3 independent experiments (****$P < 0.0001$).

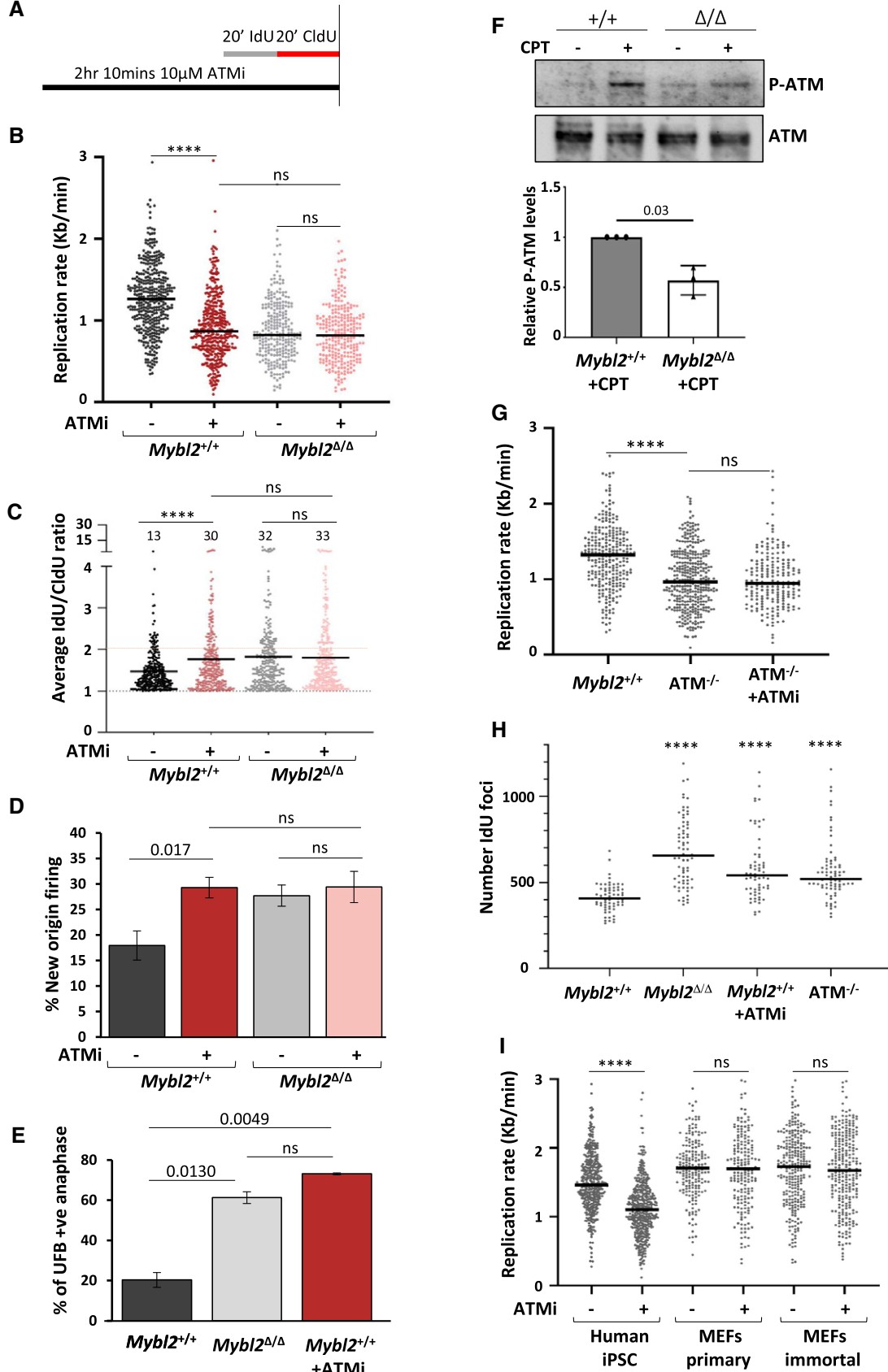

**Figure 4.**

replication stress phenotype induced by the ATM inhibitor did not arise due to off-target effects (Fig 4G). To investigate the role of ATM in suppressing replication stress further, we sought to determine whether loss of ATM activity could also lead to deregulation of replication factories. The number of replication factories in early S-phase was scored using Imaris program after cells were subjected to a short pulse of IdU. As previously reported, $Mybl2^{\Delta/\Delta}$ ESCs displayed an increase in the number of replication factories when compared to $Mybl2^{+/+}$ ESCs (Figs 4H and EV4D). In agreement with our previous observations, both $Mybl2^{+/+}$ ESCs treated with ATM inhibitor, as well as $Atm^{-/-}$ ESCs also displayed an increase in number of replication factories (Fig 4H). Altogether, these data suggest that pluripotent stem cells rely on both the ATR and ATM kinases for proper replication progression in unperturbed conditions and that MYBL2 is a key component of the ATM replication stress response pathway in ESCs.

Whilst our data are in line with a role for ATM in regulating replication in ESCs, since we have only used murine ESCs in our study, we felt that it was imperative to demonstrate that a similar phenomenon was also observed in other pluripotent stem cells but not in somatic cells. To this end, we carried out DNA fibre analysis on human induced pluripotent stem cells (iPSC) and primary/immortalized mouse embryonic fibroblasts (MEFs) in the presence or absence of an ATM, CHK2, or ATR inhibitor. Consistent with our findings using murine ESCs, the ATM, CHK2, and ATR inhibitors induced a significant reduction in replication fork speed in the human iPSCs (Fig 4I and Appendix Fig S4B and C). However, the ATM or CHK2 inhibitor did not result in a significant reduction in fork speed in the primary and immortal MEFs (Fig 4I and Appendix Fig S4B and C). Moreover, similar to what we had observed in ESCs, exposure of iPSCs to the ATM inhibitor also increased the formation of UFBs (Appendix Fig S4D). These data confirm that ATM plays a role in suppressing replication stress in pluripotent stem cells but not somatic cells.

### The replication stress phenotype in $Mybl2^{\Delta/\Delta}$ ESCs is epistatic with loss of Mre11 nuclease activity

The MRN complex (MRE11-Rad50-NBS1) is a multi-functional protein complex involved in DNA repair, DNA replication, and cell cycle checkpoint activation in part through its ability to sense DNA damage and activate the ATM/ATR-dependent DNA damage response. Previously, it has been shown that MYBL2 interacts with the MRN complex albeit the functional significance of this remains

unclear (Henrich et al, 2017). Given functional links between the MRN complex, ATM, and MYBL2, we sought to investigate whether the nucleolytic activity of MRE11, which is essential for its role in regulating DNA repair, plays a role in the MYBL2-ATM-dependent replication stress response pathway in ESCs. To test this, mirin, an inhibitor of the 3′–5′ exonuclease activity of MRE11 (Dupre et al, 2008) was utilized. $Mybl2^{+/+}$ and $Mybl2^{\Delta/\Delta}$ ESCs were treated with mirin before treatment with IdU and CldU to label nascent DNA synthesis (Fig 5A). DNA fibre analysis was performed, and the effect of mirin on replication fork rates was determined. In $Mybl2^{+/+}$ ESCs, there was a substantial decrease in replication fork rate (Fig 5B), an increase in new origin firing (Fig 5D and E) and elevated replication fork instability in response to mirin treatment (Fig 5D and F). In contrast, mirin treatment had very little impact on replication fork rates, new origin firing and replication fork stability in the $Mybl2^{\Delta/\Delta}$ ESCs (Fig 5C–F). These data indicate that both MYBL2 and the MRN complex are required for activation of the ATM-dependent replication stress response in ESCs.

### Origin firing is deregulated in $Mybl2^{\Delta/\Delta}$ ESCs and ATMi-treated $Mybl2^{+/+}$ ESCs

It is known that the DDR pathway primarily controls replication in response to DNA damage by regulating origin firing via modulation of CDK and CDC7 kinase activity (Costanzo et al, 2003; Syljuasen et al, 2005; Patil et al, 2013; Zeman & Cimprich, 2014). Deregulation of CDK activity has been demonstrated to be a major cause of replication stress partly due to aberrant origin firing (Petermann et al, 2010; Beck et al, 2012; Sorensen & Syljuasen, 2012; Anda et al, 2016). Since ATM/ATR-mediated inhibition of CDK1 is reduced in $Mybl2^{\Delta/\Delta}$ ESCs (Fig 2), we initially sought to determine whether inhibition of CDK1 or a combination of CDK2 and CDK1 could rescue the replication defect observed in $Mybl2^{\Delta/\Delta}$ ESCs and wild-type ESCs after suppression of ATM kinase activity. To investigate this, both the $Mybl2^{+/+}$ and $Mybl2^{\Delta/\Delta}$ ESCs were treated with a CDK1 inhibitor (RO3306) before labelling of active replication and DNA fibre spreading (Fig EV5A). Analysis of replication structures revealed a very mild rescue of the elevated origin firing observed in $Mybl2^{\Delta/\Delta}$ ESCs treated with CDK1 inhibitor and in $Mybl2^{+/+}$ ESCs after treatment with both CDK1 inhibitor and ATM inhibitor, although these differences were not statistically significant (Fig EV5B). Moreover, inhibition of CDK1 in the $Mybl2^{\Delta/\Delta}$ ESCs partially normalized the replication fork speed to similar levels seen in the $Mybl2^{+/+}$ ESCs after CDK1 inhibition (Fig EV5C). Lastly,

---

Figure 5. Replication speed phenotype in $Mybl2^{\Delta/\Delta}$ ESCs is epistatic with loss of MRE-11 nuclease activity.

A    Scheme of the procedure before DNA spreading. Cells were treated for 1.5 h with 20 μM MRe11 inhibitor (mirin), before sequential addition of IdU and CldU for 20 min each.

B, C    Distribution curve of replication fork rates for $Mybl2^{+/+}$ and $Mybl2^{\Delta/\Delta}$ ESCs treated with mirin. Statistical analysis was performed using the Mann–Whitney $U$-test. $n = 3$ experimental replicates. The value of the mean fork length is indicated above arrow (****$P < 0.0001$).

D    Representative images of new firing origins, and elongating forks showing stable replication and fork instability.

E    Frequency of new firing origins (relative to total structures counted) from $Mybl2^{+/+}$ and $Mybl2^{\Delta/\Delta}$ ESCs treated or not with Mirin. At least 450 replication structures were counted per treatment from three independent repeats. Error bars represent SEM. Statistical analysis using two-tailed unpaired $t$-test.

F    Distribution and the average IdU/CldU ratio in $Mybl2^{+/+}$ and $Mybl2^{\Delta/\Delta}$ ESCs with or without MRE11 inhibitor treatment. The length of both incorporated labels for each elongating fork was measured in μm, and the positive ratio was calculated. Percentage of highly unstable forks (ratio above 2) are indicated. Dotted line at 1 indicates positive ratio. $Y$-axis was cut to change scale and display large values. Statistical analysis was carried out using the Mann–Whitney $U$-test. At least 260 ratios were calculated per treatment group from 3 separate experiments (****$P < 0.0001$).

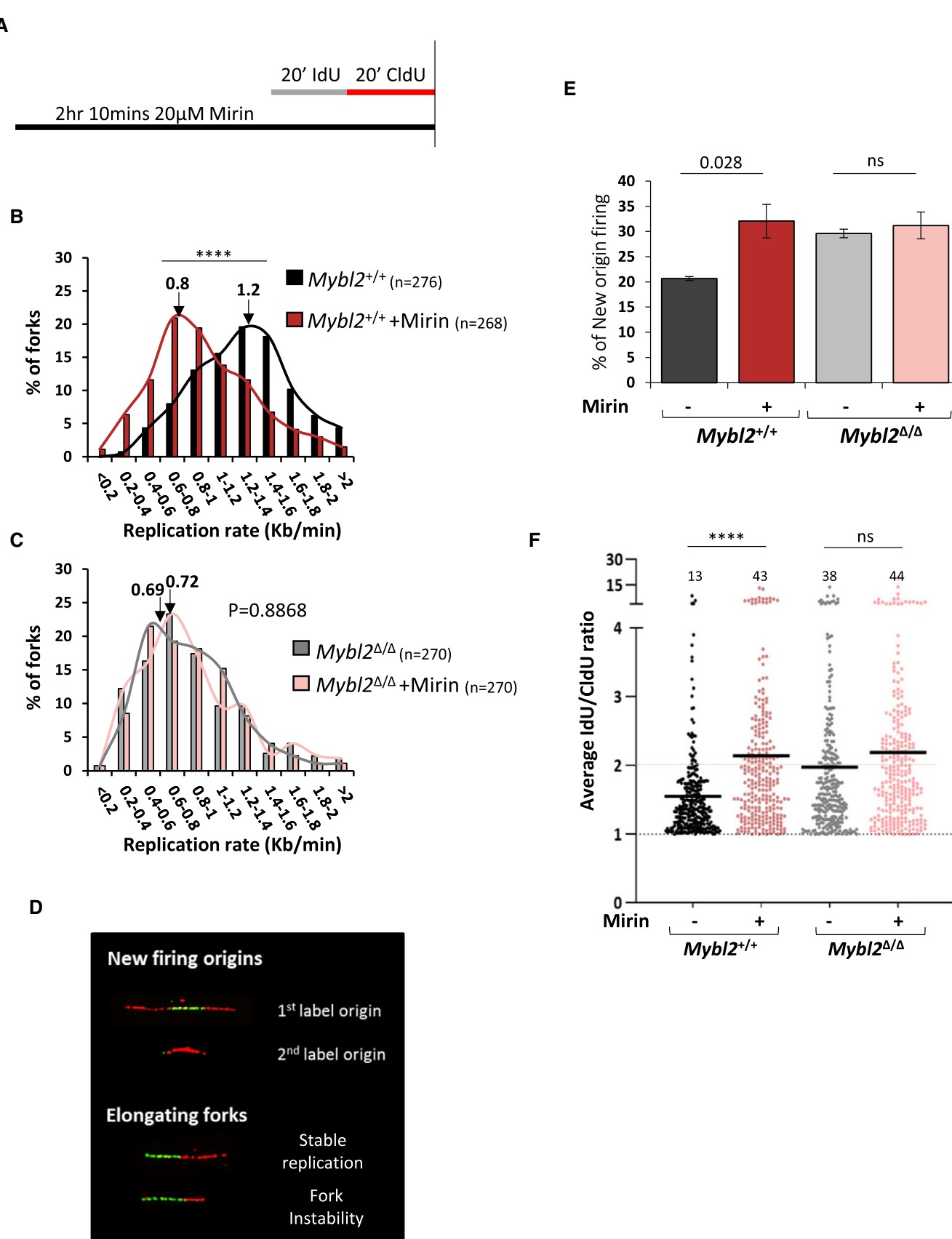

**Figure 5.**

inhibition of CDK1 in ATM inhibited ESCs also resulted in a similar partial rescue of the replication progression defect (Fig EV5C). Based on this, we reasoned that a lack of inhibition of origin firing following exposure to a CDK1 inhibitor could be due to a compensatory effect of CDK2, thus we sought to suppress origin firing by inhibiting both CDK kinases (CDK1 and CDK2) by using the CDK1/2 inhibitor III (Higgs *et al*, 2015) (Fig EV5D). Similar to the CDK1i, the CDK1/2 inhibitor also did not dramatically reduce origin firing in *Mybl2*$^{+/+}$ ESCs (neither at 3 μM nor at 25 μM) but did result in a reduced rate of replication (Fig EV5E–G). In contrast, CDK1/2 inhibition in *Mybl2*$^{\Delta/\Delta}$ ESCs and wild-type ESCs treated with an ATMi lead to a reduction in origin firing and a partial rescue of fork speed (Fig EV5E and F). From these observations, it would indicate that neither CDK1 nor CDK2 play a major role in regulating new origin firing in WT ESCs and the mild recovery of replication rates in *Mybl2*$^{\Delta/\Delta}$ ESCs following CDK1/2 inhibition may arise as a consequence of the slightly better suppression of new origin firing in these cells.

In view of these results, we hypothesized that deregulation of CDC7 may underlie the increased new origin firing and reduced replication rates exhibited by *Mybl2*$^{\Delta/\Delta}$ ESCs (Fig 6A) (Jackson *et al*, 1993; Yeeles *et al*, 2015). To test this, *Mybl2*$^{+/+}$ and *Mybl2*$^{\Delta/\Delta}$ ESCs were treated with the CDC7 inhibitor PHA-767491 (10 μM) before labelling active replication forks (Fig 6B). In agreement with this hypothesis, treatment of the *Mybl2*$^{+/+}$ ESCs with a CDC7 inhibitor resulted in a significant decrease in new origin firing (Fig 6C). Moreover, the elevated origin firing observed in the *Mybl2*$^{\Delta/\Delta}$ ESCs and in the *Mybl2*$^{+/+}$ ESCs treated with ATM inhibitor was significantly decreased to levels comparable to the *Mybl2*$^{+/+}$ ESCs (Fig 6C). Interestingly, whilst CDC7 inhibition had very little impact on replication rates in *Mybl2*$^{+/+}$ ESCs, remarkably, this resulted in a complete rescue of replication fork speed in *Mybl2*$^{\Delta/\Delta}$ ESCs or WT ESCs treated with an inhibitor (Fig 6D). In addition, inhibition of CDC7 completely rescued replication fork stability in ESCs lacking MYBL2 or ATM activity (Fig 6E). Importantly, our data indicates that the CDC7-dependent increase in origin firing was not due to an increase of CDC7 protein levels (Fig 6F). Together these data suggest that the elevated levels of replication stress resulting from a loss of MYBL2 or ATM activity is caused by aberrant CDC7-dependent firing of replication forks and that a replication stress response

pathway regulated by MYBL2 and ATM function to modulate CDC7 activity to maintain genomic integrity (Fig 7).

## Discussion

Embryonic stem cells possess unique cellular properties that are tailored to their vital roles for successful development of the mammalian embryo. Their capacity to differentiate into a diverse array of functionally distinct cell types means that the acquisition of detrimental mutations early in this process could have catastrophic consequences for the whole embryo (Blanpain *et al*, 2011). Under these circumstances, tight regulation of replication progression is paramount to maintain genomic integrity in these cells. ATR has been shown to be the principal replication stress-responsive kinase required to maintain fork stability (Paulsen & Cimprich, 2007; Cimprich & Cortez, 2008) and suppress the initiation of replication in the presence of damaged DNA (Costanzo *et al*, 2003; Syljuasen *et al*, 2005; Patil *et al*, 2013; Zeman & Cimprich, 2014). Consistent with this role, our work shows that the inhibition of ATR in both pluripotent stem cells (mouse ESCs and human iPSCs) as well as in primary and immortal somatic cells (MEFs) leads to a slow down in replication fork progression.

Importantly, our data also demonstrate a role for ATM, which is normally activated exclusively in response to DSBs (Bakkenist & Kastan, 2003; Shiloh, 2003; Lee & Paull, 2004), in regulating the replication stress response in ESCs, in part by facilitating the activation of CHK1. Whilst the mechanisms with which ATM regulates replication in ESCs have not been defined, it is clear that it does not function to suppress new origin firing and regulate elongation through the conventional CHK1-CDC25A-CDK2 DNA damage checkpoint pathway that has been identified in somatic cells (Falck *et al*, 2001). Rather, our data would indicate that CDC7 plays a more pivotal role in regulating replication origin firing than CDKs in ESCs but whether ATM directly or indirectly regulates CDC7 activation remains to be determined. Despite this, our work raises several interesting questions: Why is ATM required to regulate the replication stress response in ESCs and how does its activation in ESCs differ from that in somatic cells? It has been recently demonstrated that ESCs are very tolerant of high levels of replicative stress caused by a failure to

**Figure 6. The replication defect observed after ATM inhibition and in *Mybl2*$^{\Delta/\Delta}$ ESCs is due to deregulation of cell cycle-associated replication initiation control.**

A, B  Scheme of the aim and procedure before DNA spreading. Cells were treated for 1.5 h with ATM inhibitor (KU60019) or CDC7 inhibitor (PHA-767491) alone or in combination, before sequential addition of IdU and CldU for 20 min each.

C  Frequency of new firing origins (relative to total structures counted) from *Mybl2*$^{+/+}$ and *Mybl2*$^{\Delta/\Delta}$ ESCs treated or not with the indicated inhibitors: CDC7 inhibitor (PHA-767491) and/or ATM inhibitor (Ku60019). At least 300 replication structures were counted per treatment. Error bars represent SEM. Statistical analysis using two-tailed unpaired *t*-test (*$P < 0.05$).

D  Replication rate (kb/min) of *Mybl2*$^{+/+}$ and *Mybl2*$^{\Delta/\Delta}$ ESCs treated with the indicated inhibitors. Statistical analysis was carried out using an unpaired Mann–Whitney *U*-test (****$P < 0.0001$). Minimum three independent experiments. At least 270 forks were scored for non-ATM inhibitor-treated groups, and a minimum of 130 for ATM inhibitor-treated groups.

E  Fork stability (average IdU/CldU ratio) of *Mybl2*$^{+/+}$ ESC and *Mybl2*$^{\Delta/\Delta}$ ESCs treated with the indicated inhibitors alone or in combination. At least 240 ratios were calculated for non-ATM inhibitor-treated groups, and a minimum of 130 for ATM inhibitor-treated groups. *Y*-axis was cut to change scale and display large values. Statistical analysis was carried out using Mann–Whitney *U*-test (****$P < 0.0001$). Data for *Mybl2*$^{+/+}$ ESCs treated with ATM inhibitor alone or in combination with CDC7 inhibitor were collected from two independent experiments. For the rest of the conditions, a minimum of three independent repeats was performed.

F  Immunoblot showing the levels of CDC7 and actin in the *Mybl2*$^{+/+}$ and *Mybl2*$^{\Delta/\Delta}$ ESCs. Bar graph (lower panel) represents average band density of CDC7 in *Mybl2*$^{\Delta/\Delta}$ ESCs relative to actin and relative to *Mybl2*$^{+/+}$ ESCs from three independent repeats. Error bars represent SD. Statistical analysis was carried out using a two-tailed unpaired *t*-test (ns = no significant).

   

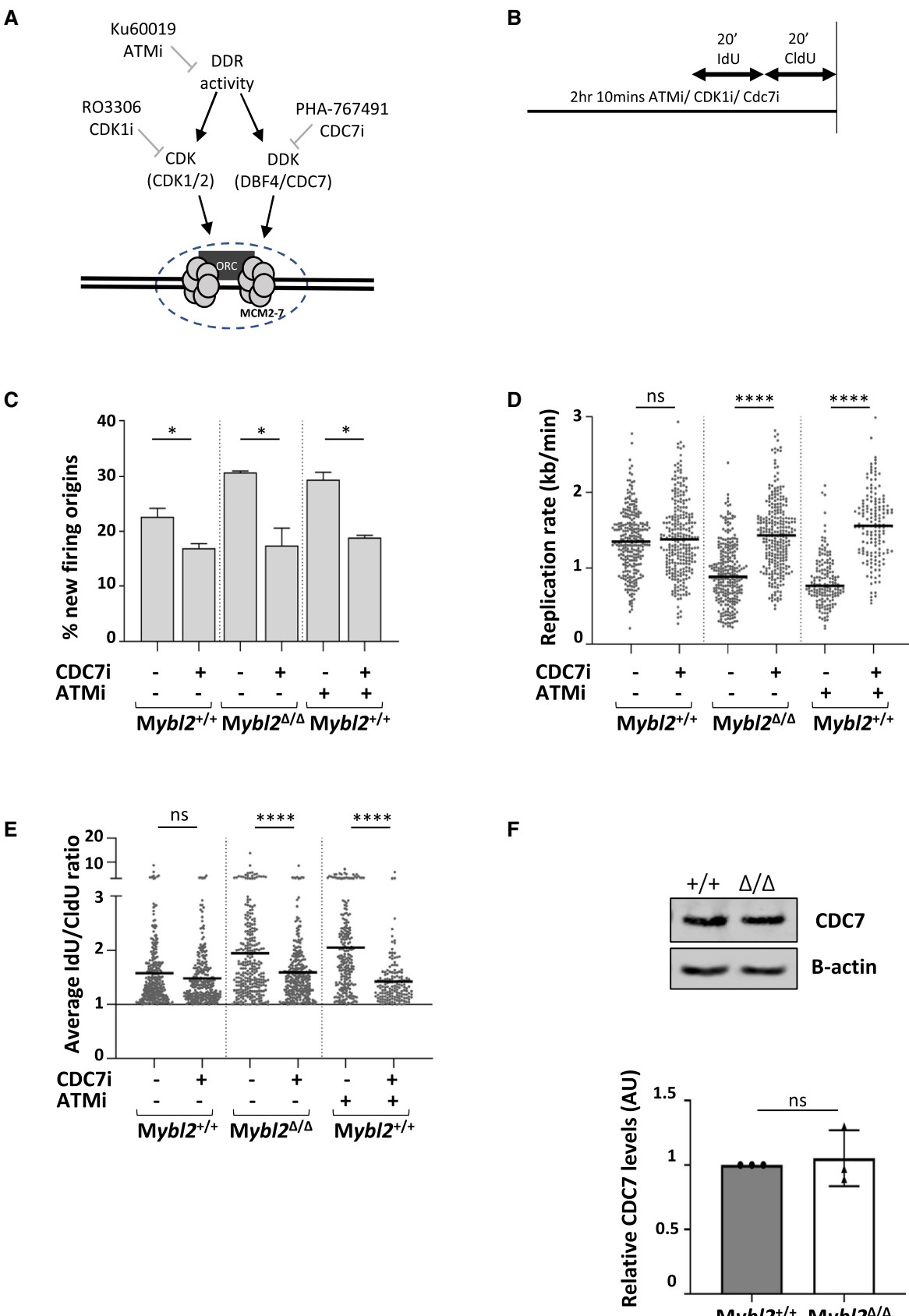

Figure 6.

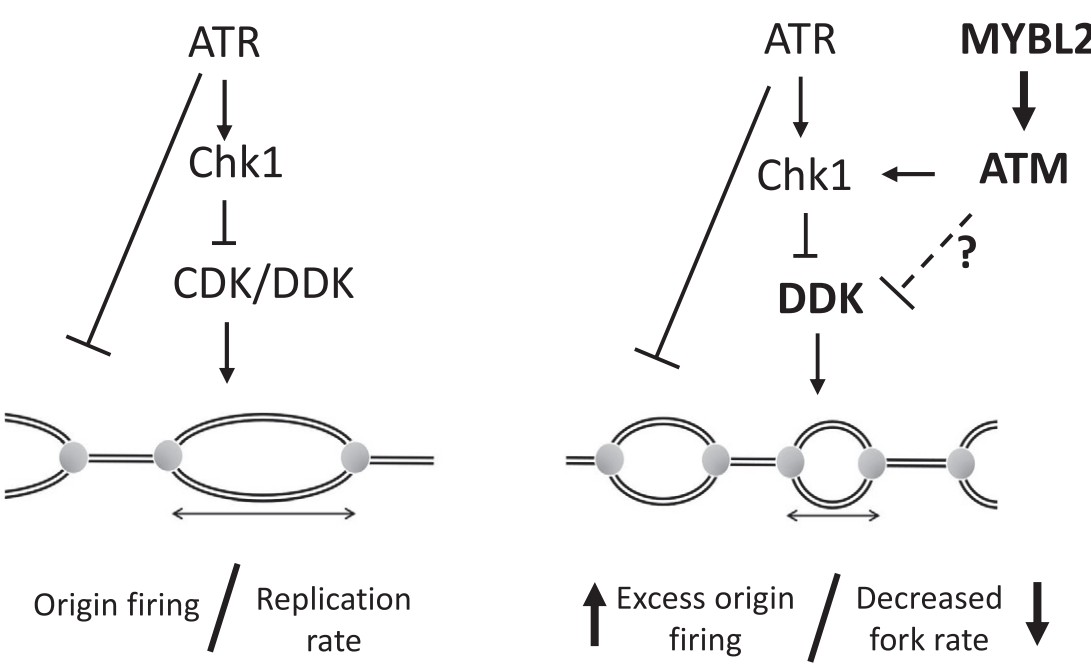

**Figure 7.  Schematic summary of our findings**

complete replication in a single cell cycle. It has been suggested that the increased propensity for replication forks in ESCs to undergo reversal coupled with their high reliance on replication coupled-repair mechanisms to deal with the under-replicated DNA allows these cells to prevent fork collapse and chromosomal breakage in the face of high levels of replication stress. Based on this, it is likely that many critical regulators of DNA replication and the replication stress response will be limiting and as a consequence, factors such as ATR require aid from back up pathways, such as those regulated by ATM. Conversely, whilst ESCs actively induce replication fork reversal to maintain genome stability from the multiple rounds of discontinuous replication, it is evident that this process is not infallible and as such, DSBs do result from replication fork collapse. Therefore, it is conceivable that this is why ESCs are more reliant on ATM to respond to DNA damage arising in S-phase than somatic cells.

In relation to whether the activation of ATM in response to replication differs in ESCs from that in somatic cells, given that DSBs are generated as a result of the unusual process of replication in ESCs and that we have observed that MRN complex is required to suppress replication stress, this would be consistent with the MRN complex sensing DSBs and then activating ATM. However, it is known that ESCs have a more relaxed chromatin state and that the strength of DDR activation in these cells is dependent on the level of chromatin compaction. Given that it has been previously shown that ATM can be activated in the absence of DSBs by alternating chromatin accessibility (Bakkenist & Kastan, 2003), it is possible that the increased chromatin accessibility in ESCs allows ATM to respond to

a wider spectrum of DNA lesions or structures than it usually would in somatic cells where regions of eu- and heterochromatin are more clearly defined. In addition to this, we have demonstrated that the activation of the MRN-ATM-dependent replication stress response in ESCs also requires MYBL2. Whilst it is not clear why MYBL2 is required to activate the ATM-dependent replication stress response, it has been previously shown that MYBL2 binds to the MRN complex and is recruited to sites of DNA damage (Henrich *et al*, 2017). However, it was suggested that MYBL2 does not play a role in regulating DNA repair and this interaction was associated with activation of the DNA damage G2/M checkpoint through its ability to act as a transcription factor (Henrich *et al*, 2017). In keeping with this report, we observed that *Mybl2*$^{\Delta/\Delta}$ ESCs failed to properly activate the DNA damage-induced G2/M checkpoint but whether this resulted from a reduction in the expression of genes involved in the G2-M transition or a failure to properly activate CHK1 remains to be determined. Nevertheless, our RNA-seq data from *Mybl2*$^{\Delta/\Delta}$ ESCs did not identify any significant alterations in genes associated with DNA replication suggesting that in ESCs MYBL2 maybe also be functioning as a component of the replication stress response independent of its role as a transcription factor.

Whilst our work clearly demonstrates that MYBL2, in conjunction with ATM, is important for suppressing replication stress in ESCs, it is currently not known what the underlying cause is. Ahuja *et al* (2016) reported that the endogenous high level of replication stress present in ESCs does not arise as a consequence of high levels of oxidative damage, deoxynucleotide (dNTP) depletion or

                                                        

increased transcription–replication collisions but was rather due to the very short G1 phase preventing sufficient time for replication stress resolution. However, from our observations, it is clear that the loss of MYBL2 or ATM significantly increases the levels of endogenous cellular replication stress in ESCs without affecting the length of G1 phase. Interestingly, we demonstrated that the high levels of replication stress exhibited by ESCs lacking MYBL2 or ATM could be rescued by a short incubation with a CDC7 inhibitor. Although it was suggested by Ahuja et al (2016) that the rescue of endogenous replication stress in ESCs following inhibition of CDC7 was solely due to a lengthening of G1 phase, this was achieved after ESCs were released from a transient G1 arrest imposed by an 8 h incubation with the CDC7 inhibitor (PHA-767491). In contrast, we could observe a rescue of replication rates in MYBL2 or ATM-deficient ESCs after a 1.5 h incubation with the same CDC7 inhibitor and at the same concentration, which is unlikely to significantly affect the duration of G1 phase.

Taken together, our work identifies ATM as a critical regulator of the replication stress response pathway in ESCs and demonstrate that MYBL2 functions to suppress replication stress, in part through its ability to activate ATM. Furthermore, we believe these findings are likely to have clinical ramifications since aberrant regulation of MYBL2 may affect the sensitivity of tumour cells to inhibitors of the ATM-dependent DNA damage response.

# Materials and Methods

### MYBL2 ablated mouse embryonic stem cells and ATM$^{-/-}$ ESCs generation and culture conditions

New $Mybl2^{\Delta/\Delta}$ ESCs were generated from $Mybl2^{F/\Delta}$ mESCs (Garcia et al, 2005) following the same protocol as previously described (Lorvellec et al, 2010). $Atm^{-/-}$ ESCs were derived from blastocysts generated from $Atm^{+/-}$ crosses as previously described (Lorvellec et al, 2010). ESCs were culture over mitomycin-treated MEFs feeder layer, using the media previously described (Ward et al, 2018). $Mybl2^{\Delta/\Delta}$ ESCs used in all experiments were not grown past 15 passages. $Atm^{-/-}$ ESCs were used before passage five. Cells were regularly tested for mycosplasma contamination.

### Inhibitors/treatments

Cells were grown under different treatments as indicated in the figures. Colcemid Gibco, catalogue number 15212012; used at 0.27 μM. Topoisomerase I inhibitor: Camptothecin (CPT), C9911, Sigma; used at 2.5–10 μM. CDK1 inhibitor: RO3306, 4181, Tocris; used at 10 μM. CDC7 inhibitor: PHA-767491, 3140, Tocris; used at 10 μM. ATM inhibitor: KU60019, CAY17502-1, Cambridge bioscience; used at 10 μM. ATR inhibitor: AZ20 5198, Tocris; used at 10 μM. CHK2 inhibitor: BML-277, Sigma; used at 10 μM. MRE-11 inhibitor: mirin M9948, Sigma; used at 20 μM.

### Western blotting

Western blotting was performed following standard procedures. Lysis buffer was previously described (Lorvellec et al, 2010). Primary antibodies: MYBL2 N19, sc724 (SCBT); P-CDK1 (Cdc2)

(Tyr15), 10A11 (Cell signalling); CDK1, A17 (Boster); P-Chk1 (Ser345), 133D3 (CST); CHK1, Sc8408 (SCBT); GADPH, Ab8245 (Abcam); B-actin, Sc1616 (SCBT); ATM 2C1 (Novus Biological); P-ATM (AF1655) GeneTex; CDC7, 3603 (Cell Signaling Technology).

### DNA fibres

Exponentially growing cells were subject to various chemical inhibitors and stress inducing agents before labelling replicating DNA through incorporation of thymidine analogues in culture. During labelling, all media was pre-heated to 37°C before adding to cells. Firstly, warm ESC media containing 30 μM iododeoxyuridine (IdU; I7125, Sigma) was added for 20 min. Cells were washed gently with warm media before addition of media containing 300 μM chlorodeoxyuridine (CldU; C6891, Sigma) for a further 20 min. Cell lysis and spreading and immunofluorescence were performed as previously described (Lorvellec et al, 2010), using Rat anti-BrdU/CldU (ab6326) and Mouse anti-BrdU/IdU (ab1816) both from Abcam. Imaging was carried out on a Leica DM6000 fluorescence microscope, and images were taken at X60 magnification. Analysis of fibres was performed using the LasX software from Leica.

### Replication factories

Embryonic stem cell colonies were exposed to experiment specific treatments before addition of 20 μM IdU (I7125; Sigma) to the media for 20 min. Immunofluorescence was performed as previously described using a mouse anti-IdU antibody (Invitrogen, MD5000) (Lorvellec et al, 2010). Image acquisition was performed using a Zeiss LSM 780 confocal immunofluorescence microscope. During the early stages of S-phase, the nucleus presents numerous small foci throughout the nucleoplasm, whereas, in mid/late S-phase cells, replication factories aggregate at the nuclear periphery before adopting a more clumped appearance (Ferreira & Carmo-Fonseca, 1997; Dimitrova & Berezney, 2002). Colonies containing early replicating cells were selected, and z-stack images were acquired at 100× magnification with a distance of 0.4 μm between each z slice. The images were then converted to 3D rendered projections using Imaris (x64.3.1; Bitplane, Zurich, Switzerland). An automatic foci counting tool within the Imaris software was then used to automatically quantify the number of foci per cell. This was through the spots function in the surpass tool. Spots were defined as "growing regions" with an estimated diameter of 0.3 μm.

### BrdU Flow cytometry

Cells were treated with 25 μM BrdU for 1 h and BrdU staining detected using FITC Mouse Anti-BrdU Set (BD Biosciences, 556028). Cells were resuspended in 10 μg/ml 7-Amino-Actinomycin D (7-AAD) (BD biosciences, 559925) for the detection of DNA content. Analysis was performed using FlowJo.

### Alkaline comet assays

$5 \times 10^4$ ESCs were used per slide following the protocol previously described (Bayley et al, 2018).

## Ultra-fine bridges

Experiment specific treatments were performed before washing cells with ice-cold PBS and fixation through treatment with 4% (v/v) PFA for 10 min. PFA was removed through PBS washes, and formaldehyde groups were quenched through treatment with 50 mM ammonium chloride for 10 min. Cells were washed three times more with PBS before adding ice-cold 100% methanol for 10 min. After repeating washes, cells were blocked in blocking buffer, 10% (v/v) FBS, 1% BSA (w/v) and 0.3% (v/v) Triton X in PBS, for 1 h at room temperature. Antibody staining was performed using an anti-ERCC6 primary antibody (H00054821, Abnova), 1 in 100 in blocking buffer at 4°C overnight, and an anti-rabbit Alexa 488 secondary antibody (A31565, Thermo Fisher) at 1 in 500 in blocking for 1 h at room temperature. Slides were mounted in prolong plus DAPI before storing at −20°C. Microscopy was performed using a Leica DM6000 fluorescence microscope; anaphase cells were visualized by DAPI staining; and the percentage of anaphases with ultra-fine bridges was determined for each sample.

## Immunofluorescences

Cells were fixed in 4% paraformaldehyde for 20 min before washing twice in PBS. Permeabilization and blocking were carried out through treatment with blocking buffer containing 1% (w/v) BSA and 0.3% (v/v) Triton X in PBS for 1 h at room temperature. Cells were then incubated with a mouse anti-H3-pS10 (9701S) primary antibody at a concentration of 1 in 200 in blocking buffer at 4°C overnight or a rabbit anti-53BP1 primary antibody (Novus Biologicals NB100-304) at 1 in 500 in blocking buffer at 4°C overnight. A separate slide was also incubated with an anti-mouse IgG antibody (Santa-cruz, sc2025) or rabbit IgG (sc-2027, Santa Cruz) for an IgG control. Slides were washed twice in PBS before incubating with goat anti-mouse Alexa 488 secondary or anti-rabbit Alexa 488 secondary antibody (A31565, Thermo Fisher). Imaging was performed using a Leica DM6000 fluorescence microscope, and images were taken at x40 magnification for 53BP1 staining or at 20× magnification for H3-pSer10 staining. The number of 53BP1 foci per nuclei/pr positive nuclei for H3-P-Ser10 was counted manually using the counting plugin for ImageJ.

For detection of H3-pSer10, prior to fixation, ESCs were treated with 2.5 μM CPT for 4 h; 0.1 μg/ml (0.27 μM) colcemid was added for the final 3.5 h to arrest cells which pass into mitosis. Cells were immediately harvested on ice before washing in PBS and centrifugation at 250 $g$ for 5 min at 4°C. $5 \times 10^4$ cells were cytospin at 300 g for 5 min onto microscope slides and air-dried for 15 min.

For detection of 53BP1 foci, 10 μM EdU (C10337, Thermo Fisher) was added to cells in culture for 1 h before harvesting on ice and a click-IT reaction, and a "click" reaction was performed after fixation and permeabilization prior to antibody staining.

## Detection of EdU through a click reaction

Staining of incorporated EdU was performed through a "click" reaction representing the reaction between an alkyne (conjugated to EdU) and an azide (fluorescently labelled). A reaction cocktail was made containing 86% Tris-buffered saline (TBS; 50 mM Tris-Cl, pH 7.5, 150 mM NaCl), 4% (v/v) 100 mM $CuSO_4$, 0.125% (v/v) Alexa-fluor azide 594 (C10330, Thermo Fisher) and 10% (v/v) 1 M sodium ascorbate added in the order presented. 200 μl of the reaction cocktail was added to each slide under a coverslip and incubated for 30 min in the dark. Slides were washed several times in PBS before re-blocking for 30 min in blocking solution.

## Data availability

No data generated in this study were deposited in public databases.

**Expanded View** for this article is available online.

## Acknowledgements

The authors wish to thank the members of the Garcia laboratory for advice and constructive criticisms. Dr Ruth Densham, Dr Clare Davies and Dr Martin Higgs for advice and critical reading of the manuscript; Dr Sovan Sarkar for the immortal MEFs and Professor Tatjana Stankovic for the $Atm^{+/-}$ mice. The authors also wish to thank the animal facility and cell sorter facility at the University of Birmingham. G.S.S is funded by a CR-UK Programme Grant (C17183/A23303). This work was also funded by a MRC PhD studentship (1632704) to D.B and an ISSF Wellcome Trust critical data award (1000599) to P.G.

## Author contributions

DB conceived and performed experiments, acquired and analysed data, and wrote the manuscript. NV-L, RA, EG, CW and MM performed experiments, acquired and analysed data. GM provided reagents. EP and AG helped with experimental design, interpretation and critical discussion of the data. GSS performed experiments, helped with experimental design, interpretation and critical discussion of the data and wrote the manuscript. PG conceived and performed experiments, acquired and analysed data, wrote the manuscript and managed the project.

## Conflict of interest

The authors declare that they have no conflict of interest.

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
