## [Review Process File · EMBO Reports]

MYBL2 and ATM suppress replication stress in pluripotent stem cells

Daniel Blakemore, Nuria Vilaplana-Lopera, Ruba Almaghrabi, Elena Gonzalez, Miriam Moya, Carl Ward, George J. Murphy, Agnieszka Gambus, Eva Petermann, Grant Stewart, and Paloma Garcia DOI: [10.15252/embr.202051120](https://doi.org/10.15252/embr.202051120)

Corresponding author(s): Paloma Garcia (p.garcia@bham.ac.uk)

Review Timeline:

Submission Date:	17th Jun 20
Editorial Decision:	14th Jul 20
Revision Received:	28th Dec 20
Editorial Decision:	5th Feb 21
Revision Received:	10th Feb 21
Accepted:	19th Feb 21

Editor: Esther Schnapp

Transaction Report:

Dear Dr. Garcia,

Thank you for the submission of your manuscript to EMBO reports. We have now received the enclosed comments from the referees.

As you will see, while the referees acknowledge that the findings are potentially interesting, they also point out that it remains unclear how ATM, MYBL2 and CDC7 are mechanistically linked. Referee 2 agrees in his/her cross-comments that additional data supporting mechanistic links between the 3 players should be provided. If you prefer, we can also discuss your potential revision plans before you start revising your study.

If you agree to embark on such a revision, I would like to invite you to revise your manuscript with the understanding that the referee concerns must be fully addressed and their suggestions taken on board. Please address all referee concerns in a complete point-by-point response. Acceptance of the manuscript will depend on a positive outcome of a second round of review. It is EMBO reports policy to allow a single round of major revision only and acceptance or rejection of the manuscript will therefore depend on the completeness of your responses included in the next, final version of the manuscript.

Revised manuscripts should be submitted within three months of a request for revision; they will otherwise be treated as new submissions. Please contact us if a 3-months time frame is not sufficient for the revisions so that we can discuss this further.

Regarding data quantification, please specify the number "n" for how many independent experiments were performed, the bars and error bars (e.g. SEM, SD) and the test used to calculate p-values in the respective figure legends. This information must be provided in the figure legends. Please also include scale bars in all microscopy images.

- 1) A data availability section providing access to data deposited in public databases is missing. If you have not deposited any data, please add a sentence to the data availability section that explains that.
- 2) Your manuscript contains statistics and error bars based on n=2 or on technical replicates. Please use scatter blots in these cases. No statistics can be calculated if n=2.

2) individual production quality figure files as .eps, .tif, .jpg (one file per figure).

See https://wol-prod-cdn.literatemonline.com/pb-assets/embo-site/EMBOPress_Figure_Guidelines_061115-1561436025777.pdf for more info on how to prepare your figures.

3) We replaced Supplementary Information with Expanded View (EV) Figures and Tables that are

collapsible/expandable online. A maximum of 5 EV Figures can be typeset. EV Figures should be cited as 'Figure EV1, Figure EV2' etc... in the text and their respective legends should be included in the main text after the legends of regular figures.

5) a complete author checklist, which you can download from our author guidelines <<https://www.embopress.org/page/journal/14693178/authorguide>>. Please insert information in the checklist that is also reflected in the manuscript. The completed author checklist will also be part of the RPF.

6) Please note that all corresponding authors are required to supply an ORCID ID for their name upon submission of a revised manuscript (<<https://orcid.org/>>). Please find instructions on how to link your ORCID ID to your account in our manuscript tracking system in our Author guidelines <<https://www.embopress.org/page/journal/14693178/authorguide#authorshipguidelines>>

7) Before submitting your revision, primary datasets produced in this study need to be deposited in an appropriate public database (see <https://www.embopress.org/page/journal/14693178/authorguide#datadeposition>). Please remember to provide a reviewer password if the datasets are not yet public. The accession numbers and database should be listed in a formal "Data Availability" section placed after Materials & Method (see also <https://www.embopress.org/page/journal/14693178/authorguide#datadeposition>). Please note that the Data Availability Section is restricted to new primary data that are part of this study. * Note - All links should resolve to a page where the data can be accessed. *
If your study has not produced novel datasets, please mention this fact in the Data Availability Section.

8) We would also encourage you to include the source data for figure panels that show essential data. Numerical data should be provided as individual .xls or .csv files (including a tab describing the data). For blots or microscopy, uncropped images should be submitted (using a zip archive if multiple images need to be supplied for one panel). Additional information on source data and instruction on how to label the files are available at <<https://www.embopress.org/page/journal/14693178/authorguide#sourcedata>>.

9) Our journal also encourages inclusion of *data citations in the reference list* to directly cite datasets that were re-used and obtained from public databases. Data citations in the article text are distinct from normal bibliographical citations and should directly link to the database records

from which the data can be accessed. In the main text, data citations are formatted as follows: "Data ref: Smith et al, 2001" or "Data ref: NCBI Sequence Read Archive PRJNA342805, 2017". In the Reference list, data citations must be labeled with "[DATASET]". A data reference must provide the database name, accession number/identifiers and a resolvable link to the landing page from which the data can be accessed at the end of the reference. Further instructions are available at <https://www.embopress.org/page/journal/14693178/authorguide#referencesformat>

I look forward to seeing a revised version of your manuscript when it is ready. Please let me know if you have questions or comments regarding the revision.

Kind regards,
Esther

Referee #1:

In this manuscript, Blackemore et al., present data for a role of Mybl2 (best known as the b-myb) and ATM in regulation of DNA synthesis in mouse embryonic stem cells (mESCs) and human induced pluripotent stem cells (iPSCs). Previous work has shown that mybl2 is involved in regulation of DNA synthesis in a variety of cell lines as well as in mESCs, through several pathways. The most innovative part of this manuscript is that inhibition of ATM in mESCs and iPSCs and not in MEFs, slows down DNA replication. This observation mirrors previous work in *Xenopus* egg extract showing that ATM negatively regulates DNA synthesis through the ATM/CDC25A/Cdk2 and Cdc7 axis, which prevents DNA replication initiation (Costanzo et al., Mol Cell 2003). Authors now propose that ATM regulates replication initiation through Mybl2 and the essential initiation kinase Cdc7, based on experiments using an ATM specific inhibitor and mybl2 knock out cells, however they did not prove how this works, whether ATM and/or Mybl2 directly regulate Cdc7 activity. No evidence for a functional interaction between ATM and Mybl2, and Mybl2 Cdc7 is provided. Because of an inefficient G1/S checkpoint, mESCs strongly rely on the ATR and ATM checkpoint to regulate cell cycle progression in the presence of DNA damage. Treatment of mESCs with Cdc7 imposes a G1/S delay (Ahuja et al., Nat Comm 2016), therefore the rescue of the replication phenotype observed in these experimental conditions could be simply due to artificial cell cycle extension. Hence, a simple

interpretation of the results reported by the authors is that by inactivating the ATM checkpoint, the endogenous DNA damage of ESCs is now detected by ATR, but higher CDK1 levels, due to Mybl2 absence, promote mitotic entry in the presence of unreplicated DNA thus increasing basal DNA damage. In this respect, authors did not test whether ATR inhibition also rescues the DNA replication slow down observed upon ATM inhibition. In conclusion, the claim that in mESCs ATM regulates DNA synthesis through the Mybl2/Cdc7 axis is not sustained by solid and mechanistic experimental evidence.

Specific points

1. Introduction, reviews related to the description of the DDR cited in the manuscript are outdated. More recent reviews must be included. References related to the mechanistic of G1/S checkpoint inefficiency in mESCs must be also included. Line 45, Ahuja et al do not show that mESCs have a robust ATR checkpoint that prevents replication of damaged DNA. They show that the high endogenous DNA damage of mESCs depends upon ATR and that actually mESCs may contain a significant fraction of unreplicated DNA (ssNDA gaps). Line 69, authors write that ATM also regulates replication timing, however the paper cited by the authors refer to ATR and not ATM.
2. In FIG. 3A, an effect of the mutant Mybl2 is seen on the phosphorylation of Chk1. It will be useful to check whether this phosphorylation comes from ATR or ATM.
3. The conclusion that the ATR pathway is partially dependent upon mybl2 is unclear from data presented in Figure 3C, the P-Chk1 level appears not to be significantly different in wild-type versus Mybl2 deleted cells.
4. It is important to show what is the cell cycle status in mESCs treated with ATM inhibitor.

Referee #2:

This is an interesting manuscript that reports a role of MYBL2 in preventing DNA replication stress in pluripotent stem cells.

Overall, the experiments are performed well and the results are clear. Specifically, it is clear that deletion of MYBL2 leads to DNA replication stress.

I would recommend publication, but also propose minor changes as follows:

1. In the absence of MYBL2, the cells have higher levels of DNA damage. This could lead to some adaptation of the DNA damage response, including a decreased checkpoint response. Therefore, the loss of MYBL2 may indirectly affect the checkpoint response. Perhaps, the authors can discuss this possibility as well.
2. In Fig 7, the authors place ATM upstream of MYBL2. However, it was not clear to me what evidence the authors have to support this placement. It could be that ATM is downstream of MYBL2. Can the authors provide some explanation why they place ATM upstream of MYBL2?
3. Is it correct to say that there is an ATM-MYBL2-CDC7 axis? It seems to me that the CDC7 inhibitor rescues replication stress, because it reduces the number of origins that fire. Is there clear evidence that MYBL2 regulates CDC7? If not, perhaps the title could focus on the epistatic relation between ATM and MYBL2.

Author's rebuttal

Referee #1: Authors now propose that ATM regulates replication initiation through Mybl2 and the essential initiation kinase Cdc7, based on experiments using an ATM specific inhibitor and mybl2 knock out cells, however they did not prove how this works, whether ATM and/or Mybl2 directly regulate Cdc7 activity. No evidence for a functional interaction between ATM and Mybl2, and Mybl2 Cdc7 is provided. Because of an inefficient G1/S checkpoint, mESCs strongly rely on the ATR and ATM checkpoint to regulate cell cycle progression in the presence of DNA damage. Treatment of mESCs with Cdc7 imposes a G1/S delay (Ahuja et al., Nat Comm 2016), therefore the rescue of the replication phenotype observed in these experimental conditions could be simply due to artificial cell cycle extension.

Response: Firstly, we would like to highlight that our inability to demonstrate a 'physical' interaction between either ATM and MMYBL2 or MYBL2 and CDC7, does not equate to our inability to provide evidence of a functional interaction between these proteins. Epistasis analysis has been used for decades to demonstrate a functional interaction between proteins and pathways. Given that we have demonstrated, using epistasis analysis, that MYBL2 functions with ATM to regulate a novel replication stress response pathway, we believe that this does constitute evidence of a 'functional' interaction between ATM and MYBL2. The fact that we can suppress the replication phenotype in ESCs lacking MYBL2 by inhibiting CDC7 provides evidence that this kinase is inappropriately regulated in the absence of MYBL2. Whether CDC7 is directly or indirectly regulated by MYBL2 does not detract from the fact that aberrant regulation of this kinase underlies the increased replication stress in ESCs lacking MYBL2. Since it is known that cells experiencing high levels of replication stress often upregulate new origin firing, we believe that the functional link between MYBL2 and CDC7 is indirect. However, this clearly warrants further investigation.

In relation to the reviewer's comments about ESCs relying more heavily on the ATM and ATR-dependent checkpoints to deal with replication stress due to having a very short G1 phase, we do not believe this is entirely correct. It is widely accepted that ESCs lack the G1/S checkpoint. Furthermore, we have demonstrated that ESCs do not appear to have a typical intra-S phase checkpoint regulated by CDK activity. Therefore, of the three known DNA damage cell cycle checkpoints, ESCs only display an ability to activate the G2/M checkpoint. Despite this, Ahuja et al. (2016) have demonstrated that WT ESCs display high levels of endogenous replication stress caused by discontinuous replication and that this does not have any impact on the ability of these cells to cycle. Thus, it would appear that the vast majority of the replication stress present in ESCs is invisible to DNA damage checkpoints. Therefore, it is likely that the reliance that ESCs have on ATM and ATR is related to their ability to stabilise stalled replication forks, protect them from uncontrolled nucleolytic degradation and to promote their repair.

Lastly, we disagree with reviewer concerning their interpretation of our use of the CDC7i. We do not believe that the addition of a CDC7i to ESCs is rescuing the replication stress phenotype merely by extending the G1/S delay. Ahuja et al. (2016) exposed ESCs to a CDC7i for 8 hours, which will certainly delay G1 cells from entering S-phase. In contrast, we added the CDC7i inhibitor for 1.5 hours and maintained it during the CldU/IdU labelling (20 min CldU and 20 min IdU). We do not believe that a 1.5 hour incubation with the CDC7i is sufficient to significantly extend the G1/S delay. Furthermore, since the CDC7i was left in with the cells during the labelling of replication forks, we are only assaying the impact of this inhibitor on cells already in S-phase. Any cells blocked by the CDC7i in G1 would not be labelled by the thymidine analogues. We have incorporated this important point within the discussion of the paper (Lines 496-512).

Referee #1: Hence, a simple interpretation of the results reported by the authors is that by inactivating the ATM checkpoint, the endogenous DNA damage of ESCs is now detected by ATR, but higher CDK1 levels, due to Mybl2 absence, promote mitotic entry in the presence of unreplicated DNA thus increasing basal DNA damage.

Response: Whilst we agree with the reviewer that it is possible that a loss of G2-M checkpoint control in ESCs lacking MYBL2 could allow DNA damage generated in S-phase to transit through mitosis undetected, thus increasing genome instability, this does not detract from the principal finding of our paper which is that MYBL2 and ATM function together in a novel replication stress response pathway in ESCs. Furthermore, we would like to add that under-replicated DNA is invisible to the DNA damage

checkpoints, such that it is unlikely that compromising this checkpoint would have any impact on this type of replication stress being passed onto daughter cells. This is evidenced by the report by Ahuja et al. (2016) showing WT ESCs have a high level of endogenous spontaneous replication stress caused by discontinuous replication. Lastly, we would like to point out that we have shown that ESCs lacking MYBL2 have similar levels of CDK1 to WT ESCs. Rather, these ESCs have reduced levels of the inhibitory phosphorylation on CDK1 (Tyr-15) catalysed by WEE1, which is probably caused, in part, by a loss of MYBL2-ATM-dependent activation of Chk1.

Referee #1: In this respect, authors did not test whether ATR inhibition also rescues the DNA replication slow down observed upon ATM inhibition. In conclusion, the claim that in mESCs ATM regulates DNA synthesis through the Mybl2/Cdc7 axis is not sustained by solid and mechanistic experimental evidence.

Response: It is not clear to us why the reviewer thinks that inhibiting the ATR-dependent replication stress response would rescue the replication stress phenotype resulting from inhibiting ATM. It has been previously shown that loss of ATR results in catastrophic levels of replication stress that are not compatible with cell viability. As such the function of ATR in embryonic cells is essential for viability. Consistent with this we have demonstrated that transient inhibition of ATR in ESCs gives rise to severe replication stress phenotype, which is much greater than we observed in ESCs lacking ATM or MYBL2. Furthermore, our recent observations would suggest that the ATM-MYBL2-dependent replication stress response pathway facilitates or serves as a partial backup for the ATR-dependent replication stress response pathway, at least with respect to promoting CHK1 phosphorylation. Based on these observations, it would seem highly unlikely that inhibiting ATR would rescue the replication stress phenotype in ESCs lacking ATM. Despite this, we have assessed replication stress using DNA fibre analysis in ESCs treated with both an ATM and ATR inhibitor. As expected the levels of replication stress induced by inhibiting both ATM and ATR are comparable to those caused by inhibiting ATR alone (Appendix Figure S4F and incorporated in the text 298-302).

Lastly, we respectively disagree that we have not supported our claims that DNA replication is maintained by an ATM-MYBL2-CDC7-dependent pathway. We have summarised our findings below:

1. We have demonstrated MYBL2 is essential for suppressing replication stress in ESCs, in part, through its ability to facilitate the activation of ATM.
2. We have demonstrated that ATM plays a novel role in suppressing replication stress in ESCs, which has not been previously shown before.
3. We have demonstrated that the increased replication stress resulting from a loss of ATM or MYBL2 is epistatic, such that this demonstrates that they function in the same novel pathway.
4. We have demonstrated that ESCs most likely lack the canonical intra-S DNA damage checkpoint that is CDK1/2 and as such rely more heavily on other mechanisms to regulate new origin firing e.g. via CDC7-dependent mechanisms
5. Lastly, we have demonstrated that suppressing new origin firing by inhibiting CDC7 can rescue the replication stress phenotype resulting from loss of MYBL2 or ATM.

Whilst we accept we have not defined whether ATM or MYBL2 regulate CDC7 directly or indirectly, this does not detract from the fact that we have identified aberrant regulation of origin firing as an underlying cause of the replication stress in ESCs lacking ATM or MYBL2. Since cells experiencing replication stress often counteract this by triggering elevated origin firing, it is not clear what is triggering this in ESCs lacking MYBL2 or ATM. Given that replication stress can be caused by multiple different stimuli, we believe that defining this is out of the scope of this current manuscript. However, we are actively pursuing this.

Reviewer 1 specific points:

1. Introduction, reviews related to the description of the DDR cited in the manuscript are outdated. More recent reviews must be included. References related to the mechanistic of G1/S checkpoint inefficiency in mESCs must be also included.

Response: We apologise for this oversight. We have now modified the introduction to incorporate the review from (Zeman and Cimprich, 2014) for the DDR section. Two references related to the

inefficient G1/S checkpoint in mESC have also been included (Stead et al, 2002; and Savatier et al, 1994).

2. Line 45, Ahuja et al do not show that mESCs have a robust ATR checkpoint that prevents replication of damaged DNA. They show that the high endogenous DNA damage of mESCs depends upon ATR and that actually mESCs may contain a significant fraction of unreplicated DNA (ssDNA gaps).

Response: This is correct, and we apologize for the over-interpretation of the results in the cited paper. We have removed this statement.

3. Line 69, authors write that ATM also regulates replication timing, however the paper cited by the authors refer to ATR and not ATM.

Response: To support this statement we have actually cited two papers. The first one by Gautier and colleagues (Shechter, NCB 2004), describes ATR and ATM in the regulation of replication timing. At the end of the discussion of this paper, the authors added a note in proof: *"While this manuscript was under review, a study by Marheineke and Hyrien reached similar conclusions (Marheineke, JBC 2004)"*.

The work of Marheineke and Hyrien describes the control of origin density and the timing of replication origin firing in *Xenopus* by using caffeine. The authors throughout the paper acknowledge that caffeine inhibits both ATR and ATM, although they decided to focus their paper in ATR. In the discussion section they reiterate this caveat of using caffeine and based on one of their experiments showing that ATR-neutralizing antibodies increase origin firing to a lower extent than caffeine, they suggest that ATM and/or other pathways also might be involved.

4. In FIG. 3A, an effect of the mutant Mybl2 is seen on the phosphorylation of Chk1. It will be useful to check whether this phosphorylation comes from ATR or ATM.

Response: We have assessed whether the increased spontaneous phosphorylation of CHK1 in the MYBL2 deficient ESCs is dependent on ATM or ATR in the absence of damage. Our data shows that this phosphorylation it is mainly ATR dependent. This result is in line with other results within the paper showing an ATM defect in MYBL2 ablated cells. (Appendix Figure S3. Lines 250-252).

5. The conclusion that the ATR pathway is partially dependent upon mybl2 is unclear from data presented in Figure 3C, the P-Chk1 level appears not to be significantly different in wild-type versus Mybl2 deleted cells.

Response: Due to inconsistent loading we have removed figure 3C. Furthermore, we have replaced the Western in Figure 3B with one that is more representative of the quantification of Chk1-P shown in this figure. Our analysis demonstrates that there is a small but significant contribution of ATM to the phosphorylation of CHK1 after the induction of DNA damage. Furthermore, whilst ESCs lacking MYBL2 do exhibit higher levels of spontaneous CHK1 phosphorylation, which we have shown is ATR-dependent, these cells cannot efficiently phosphorylate CHK1 after the induction of DNA damage induced by either CPT or IR.

However, whilst we have shown that the reduction in replication fork progression and fork stability in ESCs caused by ATR inhibition is not hugely different between WT and MYBL2 deficient ESCs (Revised Figures 3C and 3E), the increase in new origin firing is epistatic with loss of Mybl2 (Revised Figures 3D). This would indicate that MYBL2 does function within the ATR-dependent stress response pathway, albeit perhaps within a subset of functions.

Taken together, this would indicate that Mybl2 is partially epistatic with ATR.

6. It is important to show what is the cell cycle status in mESCs treated with ATM inhibitor.

Response: We have performed cell cycle experiments by flow cytometry in wild type and MYBL2-ablated ESC with and without exposure to an ATM inhibitor. The data shows that wild type and MYBL2-ablated ESC display a similar cell cycle profile, with 65-70% of the cells in S-phase. This is

consistent with what has been previously described for ESCs. Moreover, due to the short exposure of ESCs to the ATMi, we did not observe any impact of this on the ESC cell cycle profile. This data has been incorporated in Figure EV3 and in the text (Lines 297-298).

In addition to these experiments, as requested by the reviewer at a later stage, we have also:

- Expanded our fibre analysis to include immortalised MEFs to further strengthen the differences between ATM inhibition in pluripotent versus somatic cells. This new data has been incorporated into Figure 4I and referenced in Line 349-354 of the manuscript.
- Performed Western blotting for CDC7 to determine whether CDC7 protein levels are increased in MYBL2-deficient cells. This analysis did not reveal any changes in the protein levels of CDC7 resulting from a loss of MYBL2. (Figure 6F; lines 423-425).
- Performed DNA fibre analysis with inhibitor of CDK1 and CDK2 to determine whether MYBL2 and ATM regulate origin firing in general and not specifically through CDC7. (Figure EV5, lines 398-409; see outcome under reviewer 2).

Referee #2: This is an interesting manuscript that reports a role of MYBL2 in preventing DNA replication stress in pluripotent stem cells. Overall, the experiments are performed well and the results are clear. Specifically, it is clear that deletion of MYBL2 leads to DNA replication stress. I would recommend publication, but also propose minor changes as follows:

1. In the absence of MYBL2, the cells have higher levels of DNA damage. This could lead to some adaptation of the DNA damage response, including a decreased checkpoint response. Therefore, the loss of MYBL2 may indirectly affect the checkpoint response. Perhaps, the authors can discuss this possibility as well.

Response: We agree with the reviewer that the ablation of MYBL2 compromises the DNA damage checkpoint response. As shown in old Figure 2A (now figure 2C), ESCs deficient in MYBL2 have a compromised DNA damage-induced G2/M checkpoint. We also carried out DNA fibre analysis in WT and MYBL2 knockout ESCs to monitor the suppression of new origin firing following DNA damage as a marker of activation of the intra-S phase checkpoint. This analysis indicates that ESCs lack a canonical intra-S phase checkpoint. These data have been included in the main text 181-196, and in Figure 2A and 2B.

2. In Fig 7, the authors place ATM upstream of MYBL2. However, it was not clear to me what evidence the authors have to support this placement. It could be that ATM is downstream of MYBL2. Can the authors provide some explanation why they place ATM upstream of MYBL2?

Response: Whilst we felt that it was most likely that ATM functions upstream of MYBL2, given that ATM is a master controller of the DDR, we took on board this comment by the reviewer and carried out additional experiments to determine if MYBL2 is up or downstream of ATM. To address this, we determined the auto-phosphorylation status of ATM (Serine-1987 in mice) induced by DNA damage in ESCs with and without MYBL2. The auto-phosphorylation of ATM is a widely accepted marker of ATM activation. Consistent with the notion proposed by the reviewer, this analysis has revealed that ATM Ser-1987 phosphorylation is compromised in MYBL2 deficient cells after CPT treatment. This new data is consistent with MYBL2 acting upstream rather than downstream of ATM. Based on this, we have changed the title of the manuscript, the model depicted in Figure 7 and incorporated the new data in Figure 4F and Line 318-324. We sincerely thank the reviewer for this insightful comment as it has significantly changed the findings of this study and the interpretation of some of our data.

3. Is it correct to say that there is an ATM-MYBL2-CDC7 axis? It seems to me that the CDC7 inhibitor rescues replication stress, because it reduces the number of origins that fire. Is there clear evidence that MYBL2 regulates CDC7? If not, perhaps the title could focus on the epistatic relation between ATM and MYBL2.

Response: We agree with the reviewer. We don't have clear evidence for MYBL2/ATM regulating CDC7 either directly or indirectly. This needs yet to be determined; we have now mentioned this in the discussion (line 451-454). We have now performed Western blotting and shown that loss of MYBL2 does not affect CDC7 protein expression levels (Figure 6F; lines 423-425).

We have carried out additional experiments to assess whether MYBL2 binds to CDC7 directly using co-immunoprecipitation. However, these experiments were inconclusive since CDC7 runs at the same size on an SDS-Page as IgH (see below)[Figure for Referees referees not shown.].

The basis for linking ATM and MYBL2 to CDC7 relates to our ability to rescue the replication stress phenotype in ESCs lacking ATM or MYBL2 with a CDC7i. Whilst this does not formally demonstrate there is a direct signaling pathway from ATM to CDC7, it does show that CDC7 is aberrantly regulated in the absence of ATM or MYBL2. This would imply that there is a biochemical link between ATM and CDC7, albeit that this is most likely to be indirect. Based on this we will have toned down the link between ATM and CDC7, remove CDC7 from the title and modified our model accordingly.

Dear Paloma,

Thank you for the submission of your revised manuscript and for the friendly discussion today. I think your point-by-point response to referee 1's comments and concerns is fine, so please go ahead and revise your manuscript along the lines you suggest. All quantification must be provided, and please be specific when you talk about mouse or human ESC data, also in the abstract and title. If you add new data to the final manuscript, please mention this in the cover letter.

A few other more minor editorial changes will be necessary before we can proceed with the official acceptance of your manuscript.

- Please add up to 5 keywords to your manuscript file.
- A "Data Availability" section needs to be added to the end of the materials and methods and needs to state in your case, that no data generated in this study were deposited in public databases.
- Please remove the statement "data not shown" on page 15, as per journal policy. You could add/show the data if you like.
- The funding info in our online manuscript submission system and the manuscript itself does not match. Please correct.
- The quality (resolution) of the uploaded figures needs to be improved.
- The number of replicates is missing for Fig 2C, please add.
- Fig 3B has error bars that seem to be based on "at least two independent experiments". No statistics can be calculated if $n=2$, in this case, please show all datapoints of both experiments along with their mean. Or repeat the experiment at least one more time and add error bars and p-values.
- The manuscript sections are in the wrong order, please correct. The materials and methods should follow the discussion.
- In Fig EV2E, the E is a £ sign, please correct.
- I would like to suggest to change the title to something like:
MYBL2 promotes (or facilitates) ATM activation and prevents replication stress in pluripotent cells,
or
MYBL2 and ATM suppress (?) replication initiation and replication stress in pluripotent cells
- I attach to this email a related manuscript file with comments by our data editors. Please address all comments in the final manuscript.

EMBO press papers are accompanied online by A) a short (1-2 sentences) summary of the findings and their significance, B) 2-3 bullet points highlighting key results and C) a synopsis image that is exactly 550 pixels wide and 200-600 pixels high (the height is variable). You can either show a

model or key data in the synopsis image. Please note that text needs to be readable at the final size. Please send us this information along with the revised manuscript.

Referee #1:

The authors have now provided a revised version of their manuscript in which the conclusions have radically changed. Now the authors claim that MYBL2 regulates ATM activity (by an as yet unresolved molecular mechanism) and propose a new axis consisting of MYBL2-ATM-MRN as opposed to the previous conclusion in which ATM regulates MYB2L in the axis ATM-MYBL2-Cdc7. Hence, if this manuscript was to be retained in the first round of revision, it would have led to a wrong conclusion? My impression is that anything is possible. Yet, the new proposed axis remains to be entirely demonstrated. Several points I raised still remain unresolved and the citation of the appropriate literature of the G1/S checkpoint in mESCs remains poor and insufficient. In conclusion, this reinforces my concern that this work is still preliminary at the molecular level and it is not suitable for publication in EMBO Reports.

Specific points

1. Introduction. Authors are invited to read previous (and uncited) papers demonstrating that mESCs indeed activate the replication checkpoint upon exposure to replication inhibitors or DNA damaging agents (Prost et al., 1988; van der Laan et al., Mol Cell 2013; Zhao et al, Cell Res. 2018). For example, exposure of mESCs to UV-C irradiation induces a strong S-phase delay. Stead et al, 2002; and Savatier et al, 1994 do not show that mESCs do not activate the G1/S checkpoint. The reference Kapinas et al., 2013 cited in the manuscript does not refer to mouse embryonic stem cells, but to human embryonic stem cells instead. ATM was also shown to be essential in mice (reference not cited). In addition, Bakkenist et al., (Nature 2003) showed that ATM is also activated by changes in chromatin structure. Hence, the phrasing about ATM in the introduction is inconsistent.
2. Authors have not provided any evidence that the Cdc7 kinase is inappropriately regulated in mESCs in the absence of MYBL2, nor hints of how MYBL2 can regulate ATM activity. By the way, in Figure 6F, unlike what stated by the authors, it seems to me that MYBL2 knock-out appears to affect the total level of the Cdc7 protein. Because this blot was not quantified, a conclusion cannot be drawn.
3. Authors have not demonstrated that a 1.5 hour incubation with the CDC7i is not sufficient to significantly extend the G1/S delay. Further, authors have not provided any evidence that ESCs do not appear to have a typical intra-S phase checkpoint regulated by CDK activity. They have not shown the levels of CDK2/CyclinE in cells treated with CPT. Zhao et al., (Cell Res. 2018) have shown that mESCs have an increased capability to restart stalled replication forks, which is very likely what the authors score when treating cells with CPT. Authors have monitored ATM activation in cells deleted of MYBL2 (Figure 4F). They observe a small reduction in the level of phosphorylated ATM (active form), although this blot was not quantified, hence this conclusion remains a personal interpretation. This result suggests that MYBL2 may contribute to ATM phosphorylation. Yet, they

did not show whether this level is sufficient to generate a cell cycle delay (e.g.: they did not show whether phosphorylation of any ATM substrate was altered). One cannot see a strong evidence for elevated replication fork instability in response to mirin treatment (Figure 5D and 5F).

4. Appendix Figure S4F is absent

5. The schema in Figure 6A is incorrect. CDC7/Dbf4 does not affect nor binds ORC, instead it phosphorylates and affects the activity of the MCM2-7 complex.

6. In the model of Figure 7, authors propose that ATM regulates Chk1. No evidence for this result is shown and this is completely at odd with a large amount of papers showing that indeed ATM does not directly phosphorylate Chk1. However, following resection of DSBs, ATR is activated leading to Chk1 phosphorylation.

7. Discussion. Although Ahuja et al. (2016) claimed that the endogenous high level of replication stress present in ESCs is due to the very short G1 phase, this conclusion was not supported by experimental evidence, since pluripotent mESCs or mESC upon 3 days of differentiation did not show any significant difference in the length of the G1 phase, despite displaying reduced levels of gH2AX.

Referee #2:

My comments for the original version were minor. The authors have addressed them and, in fact, modified their conclusions, so that they are now fully supported by the data. The manuscript represents a significant body of work and addresses a very important question: how ESCs maintain genomic stability. I think the manuscript can be published essentially as is.

Cross-comments by referee 2:

It is true that the authors changed their model, based on two points I raised during the review of the original manuscript.

This can be viewed positively (they are not stuck on a specific model and are willing to change it, if the data indicate that this is the case) or negatively (they will propose any model, as long as the paper is published and they don't care if the model is correct). I am leaning towards the positive interpretation, but it is true that one would like to have very strong evidence supporting a specific model.

In my opinion the authors present some interesting data on the effect of MYBL2 on the response of ESCs to CPT, but mechanistically the manuscript could be stronger.

Reviewer 1 has some valid points, which I missed. I would therefore ask the authors to revise the manuscript once more, addressing all the comments of reviewer 1 by modifying the text, etc and to only do experiments to place MYBL2 upstream of ATM. If they cannot provide more experiments on this point, then the title of the manuscript could be: "MYBL2 and ATM prevent replication stress in pluripotent cells"

Overall, I think that this manuscript can be published in EMBO Reports with the experiments already performed. It is the interpretation that needs to be changed, if they cannot provide more experiments.

The authors have now provided a revised version of their manuscript in which the conclusions have radically changed. Now the authors claim that MYBL2 regulates ATM activity (by an as yet unresolved molecular mechanism) and propose a new axis consisting of MYBL2-ATM-MRN as opposed to the previous conclusion in which ATM regulates MYBL2 in the axis ATM-MYBL2-Cdc7. Hence, if this manuscript was to be retained in the first round of revision, it would have led to a wrong conclusion? My impression is that anything is possible. Yet, the new proposed axis remains to be entirely demonstrated.

Response: Respectfully, we would like to point out that the manuscript has NOT changed radically. The primary conclusions of our manuscript remain exactly the same i.e. ATM plays an unexpected role in regulating the replication stress response in ESC, which is contrary to the case in somatic cells, where this response is controlled by ATR, and that MYBL2 and ATM function together.

The ONLY difference between the first and second versions of our manuscript is where we placed MYBL2 in our hypothetical model. Given that ATM (and ATR) are master controllers of the DNA damage response, which is mediated through their ability to regulate numerous substrates by phosphorylation, including a number of transcription factors, it seemed logical that the epistasis that we observed between the loss of MYBL2 and ATM, was due to MYBL2 functioning downstream of ATM. In the manuscript, we indicated that our model was hypothetical and based on the information we had at the time. Based on comments from reviewers that we needed to obtain more insight as to the nature of the relationship between MYBL2 and ATM, we carried out further experiments. These experiments involved assessing the activation of ATM after exposure to CPT, using the phospho-S1987-ATM antibody (a known marker of ATM activation) in cells with and without MYBL2. We fully expected that a loss of MYBL2 would not affect ATM auto-phosphorylation. However, as with all scientific experiments, often the outcome you were expecting is not always the one that you get. Whilst this new data suggested that MYBL2 may function upstream of ATM (at least in ESCs), it does not detract from the overall conclusions of the manuscript.

Based on this, we feel that the reviewer is unjustified in stating that 'if this manuscript was to be retained in the first round of revision, it would have led to a wrong conclusion'. Hypothetical models as the name suggests are hypothetical and therefore subject to change. Since the changes that we have made to this model were in line with direction from the reviewers, it seems unfair to criticize us if the model changes based on new data. The whole point of the review process is to strengthen a manuscript by carrying out additional experiments to address scientific weaknesses.

As such, we believe that the revision has made the manuscript stronger. The presence of a stem cell specific regulator of ATM activation would help to explain why ATM functions in a completely new DNA damage response pathway, which is not present in somatic cells. It seems unreasonable to expect us to completely delineate the fine mechanistic details of a new pathway in a single manuscript. With this in mind, we would like to highlight that one of the first manuscripts documenting a role for the Mre11-Rad50-Nbs1 (MRN) complex regulating the activation of ATM in somatic cells published in EMBOJ (Uziel et al., 2003, 22:5612-5621) did not provide any such mechanistic insight before being published. Moreover, it has taken years of detailed research by many laboratories to ascertain how exactly the MRN complex activates ATM. Despite this, we feel that our demonstration that: 1) a loss of MYBL2 compromises the replication stress response in ESCs and is epistatic with the loss of ATM or the inhibition of the nuclease activity of Mre11, 2) a loss of MYBL2 reduces the ability of ATM to be activated, 3) a loss of MYBL2 affects the ability of Chk1 to be phosphorylated efficiently, and 4) the inhibition of Chk2 also induces replication stress, which is epistatic with loss of MYBL2 (additional new data Appendix Figure S4A and S4B), is mechanistic detail enough to support the conclusions of the manuscript.

Several points I raised still remain unresolved and the citation of the appropriate literature of the G1/S checkpoint in mESCs remains poor and insufficient. In conclusion, this reinforces my concern that this work is still preliminary at the molecular level and it is not suitable for publication in EMBO Reports.

Again we respectfully disagree with the reviewer. The experiments performed were discussed and agreed with the editor. During conversations it was reiterated by the editor that not all the experiments were needed but we had to try to clarify whether CDC7 was within the MYBL2-ATM replication stress axis, which we had previously suggested, and possibly try and shed some light on the relationship between MYBL2 and ATM. Given the severe restrictions due to COVID19 that have been placed on us by the Government about the number of personnel allowed in the laboratory at any one time and the limited number of antibodies and other reagents commercially available to study the DNA damage response in mouse cells, we believe that we have addressed the major concerns raised by the reviewers and agreed with the editor to the best of our ability.

In addition, whether our manuscript has not referenced all the current literature pertaining to whether ESCs retain or not a G1/S checkpoint in ESCs, is not a legitimate basis for concluding that our research is too preliminary. It is clear that this issue is rather controversial in the field, but we feel that this does not have any relevance to the conclusions of our manuscript. We have focused on the presence or absence of an intact intra-S and G2/M phase DNA damage checkpoint in ESCs and how the activation of these is affected by loss of ATM and MYBL2. Despite this, we

Rebuttal February

performed cell cycle analysis on ESCs in the presence or absence of an ATM inhibitor to determine if a shortening/elongation of G1 phase could explain any of our phenotypes, as reviewer 1 requested. Since we observed no difference in the cell cycle profile of WT or MYBL2 deficient ESCs in the presence/absence of an ATM inhibitor, we concluded that the issue surrounding the presence of an intact G1/S checkpoint in ESCs is not relevant to our story.

We are more than happy to include some of the reviewer's suggested references relating to the G1/S checkpoint in ESCs and also tone down any statements pertaining a 'lack' of this checkpoint.

1. Introduction. Authors are invited to read previous (and uncited) papers demonstrating that mESCs indeed activate the replication checkpoint upon exposure to replication inhibitors or DNA damaging agents (Prost et al., 1988; van der Laan et al., Mol Cell 2013; Zhao et al, Cell Res. 2018). For example, exposure of mESCs to UV-C irradiation induces a strong S-phase delay.

Response: Following reading the papers suggested by the reviewers, it is not clear whether ESCs actually possess a fully intact replication checkpoint:

Observations from Prost et al., 1998: UV irradiation can transiently slow DNA replication (as judged by BrdU incorporation and flow cytometry) in ESCs. However, there is absolutely no analysis of checkpoint activation.

Observations from van der Laan et al., 2013: UV has a very mild effect on DNA replication progression (as judged by BrdU incorporation and flow cytometry). This is in stark contrast to the complete block of replication in somatic cells. ESCs have high levels of CDC25A (which is the main target of Chk1), which is not fully degraded following UV irradiation. This suggests that the replication checkpoint in ESCs is not completely active.

Observations from Zhao et al., 2018: A direct quote is taken from the paper in relation to new origin firing following HU treatment in ESCs - 'more dormant replication origins were fired in ESCs than in the other cell types in response to HU treatment (Figure 1D), consistent with a recent report'. This observation is consistent with our findings and is indicative of an inability of ATR (and/or ATM) to suppress new origin firing in response to replication stress; a known hallmark of activation of the intra-S phase checkpoint.

Based on this, it is unclear to us what point the reviewer is trying to raise with respect to these manuscripts and their relationship to ours.

Stead et al, 2002; and Savatier et al, 1994 do not show that mESCs do not activate the G1/S checkpoint.

Response: The two references Stead et al., 2002, and Savatier et al., 1994, were cited for “weak G1/S checkpoint”. In our introduction there is no mention of these two references related to activation of G1/S checkpoint. Stead et al., 2002 demonstrates a high level of CDK2-CycE/A expression in ESCs as well as the lack of CDK inhibitors. Savatier et al., 1994 demonstrate that ESCs exhibit high levels of inactive RB1 protein. Taken together, these observations represent a likely underlying cause of the weak G1/S checkpoint observed in ESCs. Furthermore, these papers have been cited in many reviews relating to the weak G1/S checkpoint in embryonic stem cells.

The reference Kapinas et al., 2013 cited in the manuscript does not refer to mouse embryonic stem cells, but to human embryonic stem cells instead.

Response: Indeed, this is correct. However, the paragraph in our introduction is about all types of ESCs and their cell cycle peculiarities i.e. highly proliferative and short GAP phases, which are common to mouse and human. At this stage we do not make any distinctions between mouse or human ESCs. We have now clarified throughout the introduction whether we are referring to mouse or/and human ESCs.

ATM was also shown to be essential in mice (reference not cited).

Response: We respectively disagree with the reviewer. A complete loss of Atm is not embryonic lethal in mice. There are several published mouse models of Atm loss. Furthermore, the Atm null ESCs used in this manuscript were derived from an Atm null mouse model obtained from the Wynshaw-Boris lab (Barlow et al., 1996 Cell 86:159-171). It is possible that the reviewer is getting confused with the fact that the Atm kinase dead knock-in mouse is lethal. This is most likely because the presence of a catalytically dead protein is having a dominant negative effect on processes downstream of Atm relating to the processing and repair of DSBs.

In addition, Bakkenist et al., (Nature 2003) showed that ATM is also activated by changes in chromatin structure. Hence, the phrasing about ATM in the introduction is inconsistent.

Response: We apologise for this oversight. This is a mistake generated while restructuring the introduction. We have removed this citation that was duplicated in the wrong paragraph.

2. Authors have not provided any evidence that the Cdc7 kinase is inappropriately regulated in mESCs in the absence of MYBL2, nor hints of how MYBL2 can regulate ATM activity.

Response: Whilst it is possible that a loss of MYBL2 may specifically deregulate CDC7 activity and that this is contributing to the replication stress phenotype observed in MYBL2 cells, it is also possible that the replication stress caused by loss of MYBL2 is causing the cell to respond by increasing the firing of new origins to facilitate the completion of DNA replication.

We investigated the possibility that MYBL2 might be directly associating with CDC7 using co-immunoprecipitation experiments but unfortunately these experiments were inconclusive and confused by CDC7 running on a Western at the same size IgH. Despite this, our observations that reducing CDC7 activity (but not CDK activity) could reduce the replication stress in MYBL2 deficient cells clearly indicates that CDC7-dependent origin firing (whether this being directly or indirectly due to loss of MYBL2-dependent regulation) contributes to this phenotype. These experiments were performed in agreement with the editor to clarify a point made by reviewer 2, although he/she did not explicitly ask for it. .

In respect to the relationship between ATM and MYBL2, as stated above, we have demonstrated: 1) a loss of MYBL2 compromises the replication stress response in ESCs and is epistatic with the loss of ATM or the inhibition of the nuclease activity of Mre11, 2) a loss of MYBL2 reduces the ability of ATM to be activated, 3) a loss of MYBL2 affects the ability of Chk1 to be phosphorylated efficiently, and 4) the inhibition of Chk2 also induces replication stress, which is epistatic with loss of MYBL2 (additional new data Appendix Figure S4A and S4B), is mechanistic detail enough to support the conclusions of the manuscript.

Additional we did attempt to investigate whether ATM and MYBL2 could interact using co-immunoprecipitation but again the results from these experiments were inconclusive. Furthermore, these experiments are complicated by the lack of reagents that detect/immunoprecipitate mouse ATM and MYBL2.

By the way, in Figure 6F, unlike what stated by the authors, it seems to me that MYBL2 knock-out appears to affect the total level of the Cdc7 protein. Because this blot was not quantified, a conclusion cannot be drawn.

Response: As requested by the reviewer, we have quantified the levels of CDC7 in WT and MYBL2 deficient ESCs and there is no difference. The densitometry has been included in the Figure 6F.

3. Authors have not demonstrated that a 1.5 hour incubation with the CDC7i is not sufficient to significantly extend the G1/S delay.

Response: The rationale behind doing this experiment for the reviewer was: *“Treatment of mESCs with Cdc7 imposes a G1/S delay (Ahuja et al., Nat Comm 2016), therefore the rescue of the replication phenotype observed in these experimental conditions could be simply due to artificial cell cycle extension”*.

As highlighted in our previous rebuttal we added the CDC7i inhibitor for 1.5 hours and maintained it during the CldU/IdU labelling (20 min CldU and 20 min IdU). We do not believe that a 1.5 hour incubation with the CDC7i is sufficient to significantly extend the G1/S delay. Furthermore, since the CDC7i was left in with the cells during the labelling of replication forks, we are only assaying the impact of this inhibitor on cells already in S-phase. Any cells blocked by the CDC7i in G1 would not be labelled by the thymidine analogues.

Further, authors have not provided any evidence that ESCs do not appear to have atypical intra-S phase checkpoint regulated by CDK activity. They have not shown the levels of CDK2/CyclinE in cells treated with CPT. Zhao et al., (Cell Res. 2018) have shown that mESCs have an increased capability to restart stalled replication forks, which is very likely what the authors score when treating cells with CPT.

Response: Our evidence about the presence of an atypical intra-S-phase checkpoint in ESCs is based on our inability to observe a suppression of new origin firing inhibition when WT ESC are treated with either CPT or CDK inhibitors. This observation is actually consistent with one of the papers highlighted by the reviewer: **Van der Laan 2013**, in which they showed that after UV irradiation, ESCs do not efficiently suppress DNA replication as compared to NIH3T3s.

Whilst we do not disagree with the reviewer about their comments pertaining to whether ESCs can restart stalled replication forks following HU exposure, our analysis was carried out using CPT, which causes replication stress in a very different way to HU. HU stalls all replication forks by inhibiting RNR and reducing dNTP pools. CPT induces DNA-protein cross-links adjacent to a single-strand break. Depending on the dose of CPT, this can induce replication fork reversal and/or fork collapse and a one-end DNA double strand break. Since the CPT was added during the labelling procedure, our analysis does not measure replication fork restart and merely assesses the ability of cells to replicate in the presence of a CPT-induced lesion. Therefore, it is difficult to see how the results from Zhao et al., relate to our findings.

Authors have monitored ATM activation in cells deleted of MYBL2 (Figure 4F). They observe a small reduction in the level of phosphorylated ATM (active form), although this blot was not quantified, hence this conclusion remains a personal interpretation. This result suggests that MYBL2 may contribute to ATM phosphorylation. Yet, they did not show whether this level is sufficient to generate a cell cycle delay (e.g.: they did not show whether phosphorylation of any ATM substrate was altered). One cannot see a strong evidence for elevated replication fork instability in response to mirin treatment (Figure 5D and 5F).

Response: We have quantified the level of ATM phosphorylation induced by CPT in WT and MYBL2 deficient ESCs and it is significantly reduced. We have included the densitometry in the Figure 4F.

Furthermore, we have shown that G2/M DNA damage checkpoint, which is known to be regulated by ATM, is defective in MYBL2 deficient ESCs. In addition to this, we have previously demonstrated that the phosphorylation of KAP1, a known substrate of ATM, in response to ionising radiation is reduced in MYBL2 deficient hematopoietic stem cells (Bayley et al., 2018 Cancer Research).

In relation to Figure 5D and 5F, we strongly disagree with the reviewer. In Figure 5F, we demonstrate that Mirin induces a striking increase in replication fork instability in WT ESCs that has a 4 star statistical significance. We also demonstrate that the increased replication fork instability induced by Mirin is epistatic with a loss of MYBL2. This reinforces our observations that MYBL2 functions with ATM to suppress replication stress in ESCs.

5. The schema in Figure 6A is incorrect. CDC7/Dbf4 does not affect nor binds ORC, instead it phosphorylates and affects the activity of the MCM2-7 complex.

Response: We were not trying to indicate that CDC7 binds ORC, but regulating origin firing activating the replisome. We will add a circle to indicate replisome or move arrow to MCMs.

6. In the model of Figure 7, authors propose that ATM regulates Chk1. No evidence for this result is shown and this is completely at odd with a large amount of papers showing that indeed ATM does not directly phosphorylate Chk1. However, following resection of DSBs, ATR is activated leading to Chk1 phosphorylation.

Response: In Figure 3B, we demonstrate that the inhibition of ATM decreases the phosphorylation of Chk1 in WT ESCs but not in MYBL2 deficient cells.

Whilst there is a consensus in the field that Chk1 is a specific substrate of ATR and Chk2 is a specific substrate of ATM. In reality, it is not so black and white. There are certainly published paper documenting that ATM can regulate Chk1 phosphorylation. This has been reviewed by Bartek and Lukas, Cancer Cell, 2003 (scheme with figure 1B below showing that this is the case). We have attached a quote from this review:

“The original concept of rather strict dependency of Chk1 on ATR, and Chk2 on ATM, has recently been softened by reports of various “crosstalks” among these kinases, exemplified by phosphorylation/activation of Chk1 by ATM in response to ionizing radiation (Gatei et al., 2003, Sørensen et al., 2003), the identification of a novel checkpoint cascade signaling via ATM-Chk1 to Tlk kinases and thereby likely to chromatin remodeling in response to various stresses (Groth et al., 2003), reports of ATM-independent activation of Chk2 (Hirao et al., 2002), as well as by the ATX kinase whose links to Chk1 and Chk2 remain to be elucidated”.

Furthermore, we would like to add that the vast majority of the data assessing the role of ATR and ATM in activating Chk1 and Chk2 has been carried out in somatic cells and not ESCs. From our work, it is clear that ATM plays a more prominent role in the replication stress response in ESCs than in somatic cells and as such, it would make sense that the ability of ATM to regulate Chk1 in ESCs would be enhanced.

Discussion. Although Ahuja et al. (2016) claimed that the endogenous high level of replication stress present in ESCs is due to the very short G1 phase, this conclusion was not supported by experimental evidence, since pluripotent mESCs or mESCs upon 3 days of differentiation did not show any significant difference in the length of the G1 phase, despite displaying reduced levels of gH2AX.

Response: It is unclear to us why the reviewer is using a personal criticism of the paper by Ahuja et al., as a criticism of our paper, especially when the reviewer initially criticised our findings in favour of the work published by Ahuja et al: *“Treatment of mESCs with Cdc7 imposes a G1/S delay (Ahuja et al., Nat Comm 2016), therefore the rescue of the replication phenotype observed in these experimental conditions could be simply due to artificial cell cycle extension”*

Since we have demonstrated that loss of MYBL2 or ATM has little impact on the cell cycle profile of ESCs, we cannot see the relevance of this comment to our work.

Dr. Paloma Garcia
Institute of Biomedical Research
Immunity and Infection
University of Birmingham
Vincent Drive
Birmingham B15 2TT
United Kingdom

Dear Paloma,

I am very pleased to accept your manuscript for publication in the next available issue of EMBO reports. Thank you for your contribution to our journal.

At the end of this email I include important information about how to proceed. Please ensure that you take the time to read the information and complete and return the necessary forms to allow us to publish your manuscript as quickly as possible.

As part of the EMBO publication's Transparent Editorial Process, EMBO reports publishes online a Review Process File to accompany accepted manuscripts. As you are aware, this File will be published in conjunction with your paper and will include the referee reports, your point-by-point response and all pertinent correspondence relating to the manuscript.

If you do NOT want this File to be published, please inform the editorial office within 2 days, if you have not done so already, otherwise the File will be published by default [contact: emboreports@embo.org]. If you do opt out, the Review Process File link will point to the following statement: "No Review Process File is available with this article, as the authors have chosen not to make the review process public in this case."

Should you be planning a Press Release on your article, please get in contact with emboreports@wiley.com as early as possible, in order to coordinate publication and release dates.

Thank you again for your contribution to EMBO reports and congratulations on a successful publication. Please consider us again in the future for your most exciting work.

THINGS TO DO NOW:

You will receive proofs by e-mail approximately 2-3 weeks after all relevant files have been sent to our Production Office; you should return your corrections within 2 days of receiving the proofs.

Please inform us if there is likely to be any difficulty in reaching you at the above address at that time. Failure to meet our deadlines may result in a delay of publication, or publication without your corrections.

All further communications concerning your paper should quote reference number EMBOR-2020-51120V3 and be addressed to emboreports@wiley.com.

Should you be planning a Press Release on your article, please get in contact with emboreports@wiley.com as early as possible, in order to coordinate publication and release dates.

Corresponding Author Name: PALOMA GARCIA

Journal Submitted to: EMBO REPORTS

Manuscript Number: EMBOR-2020-51120V1